# Causal networks of phytoplankton diversity and biomass are modulated by environmental context

Chun-Wei Chang[1,2], Takeshi Miki[3,4,5], Hao Ye[6], Sami Souissi[7], Rita Adrian[8,9], Orlane Anneville[10], Helen Agasild[11], Syuhei Ban[12], Yaron Be'eri-Shlevin[13], Yin-Ru Chiang[14], Heidrun Feuchtmayr[15], Gideon Gal[13], Satoshi Ichise[16], Maiko Kagami[17,18], Michio Kumagai[16,19], Xin Liu[12], Shin-Ichiro S. Matsuzaki[20], Marina M. Manca[21], Peeter Nõges[11], Roberta Piscia[21], Michela Rogora[21], Fuh-Kwo Shiah[2,4], Stephen J. Thackeray[15], Claire E. Widdicombe[22], Jiunn-Tzong Wu[14], Tamar Zohary[13] & Chih-hao Hsieh[1,2,4,23 ✉]

Untangling causal links and feedbacks among biodiversity, ecosystem functioning, and environmental factors is challenging due to their complex and context-dependent interactions (e.g., a nutrient-dependent relationship between diversity and biomass). Consequently, studies that only consider separable, unidirectional effects can produce divergent conclusions and equivocal ecological implications. To address this complexity, we use empirical dynamic modeling to assemble causal networks for 19 natural aquatic ecosystems (N24°~N58°) and quantified strengths of feedbacks among phytoplankton diversity, phytoplankton biomass, and environmental factors. Through a cross-system comparison, we identify macroecological patterns; in more diverse, oligotrophic ecosystems, biodiversity effects are more important than environmental effects (nutrients and temperature) as drivers of biomass. Furthermore, feedback strengths vary with productivity. In warm, productive systems, strong nitrate-mediated feedbacks usually prevail, whereas there are strong, phosphate-mediated feedbacks in cold, less productive systems. Our findings, based on recovered feedbacks, highlight the importance of a network view in future ecosystem management.

A full list of author affiliations appears at the end of the paper.

Since ecosystems were first described as delicate feedback systems by Tansley[1], feedback has been a recurring theme among global-scale ecosystem studies[2–5]. A *feedback* is defined as a directed and connected path of causal interactions that ends on the originating node (i.e., a "cycle" in network terminology). As feedbacks have a critical role in dynamical systems over long-term observations[6,7], feedbacks have been suggested as an important consideration for elucidating interactions between biodiversity and ecosystem functioning (BDEF)[8] and how they regulate natural systems[9]. However, biodiversity and ecosystem functioning are only two components of a larger interconnected network with many causal links and feedbacks[10–12] among a multitude of environmental factors, including nutrient availability[13] and temperature[14]. Ignoring these feedbacks and the role of environmental factors can complicate the interpretation of BDEF relationships. For example, impacts of plant diversity loss on plant biomass cannot be precisely evaluated if feedbacks among plant diversity, biomass, and environment are overlooked[8].

A holistic view of the causal network associated with biodiversity, integrating posited relationships from previous studies, is shown (Fig. 1). Here, each arrow represents a simple causal interaction (e.g., BD → EF depicts biodiversity effects on ecosystem functioning). In addition to pairwise feedback between biodiversity and ecosystem functioning (BD ↔ EF), more complex triangular feedbacks exist when including *endogenous* environmental factors (e.g., nutrients) that can affect and be affected by organisms[15]. Considering this complexity in natural ecosystems, we need to incorporate feedbacks into the current research framework of biodiversity.

Although causal networks among biodiversity, ecosystem functioning, and the environment have been discussed[10–12],

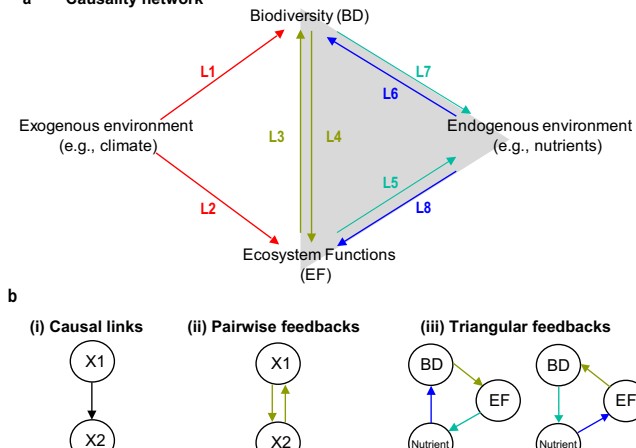

**a** Causality network

**b**

(i) Causal links   (ii) Pairwise feedbacks   (iii) Triangular feedbacks

**Fig. 1 Conceptual causal network of biodiversity and ecosystem functioning (BDEF) relationships.** Environmental variables in the causal network (**a**) can be exogenous (e.g., climate), which influence BD and EF, or endogenous (e.g., nutrients), which influence and can be influenced by BD and EF. Whereas endogenous factors can affect and be affected by organisms[5], exogenous factors, such as precipitation and temperature, can only affect ecosystems (organisms do not influence precipitation and temperature on the scales considered in a majority of ecological studies, e.g., daily, monthly, or annual scales) and therefore cannot be included in feedbacks[26]. The causal network can be decomposed into modules (**b**): (i) individual causal links (e.g., L1-L8), (ii) pairwise feedbacks (e.g., L3-L4), and more complex (iii) triangular feedbacks. Triangular feedbacks connecting biodiversity, ecosystem functioning, and nutrients (gray triangle in (**a**)) can be classified based on direction: BD → EF → Nutrients (Type I feedback: L4-L5-L6) and EF → BD → Nutrients (Type II feedback: L3-L7-L8).

quantification of these networks is yet to be fully realized. Because manipulating multiple interdependent processes is infeasible[12,16], experimental studies often examine individual interactions in isolation[17]. As previously noted, however, these studies might ignore the context of other relevant factors[18]. Alternatively, empirical reconstructions of causal networks from observational data have often led to equivocal results[19,20]. The use of linear statistical methods may be an explanation: correlation and regression-based approaches assume static relationships among variables and are not designed to investigate interdependent feedbacks that produce time-varying interactions observed in natural systems[21–23]. Consequently, there is a lack of network-based approaches that can identify relative contributions of various drivers to a focal ecosystem process and explore conditions under which their contributions might change. For instance, diversity is recognized as the strongest determinant for ecosystem functioning[24] in experimental systems (i.e., L4 in Fig. 1). However, in natural systems, debate remains over whether the effect of diversity on ecosystem functioning is stronger than exogenous (L2 in Fig. 1) or endogenous drivers[25] (L8 in Fig. 1), both of which affect organisms[5], although only endogenous drivers can be affected by organisms and involved in feedbacks[26]. Similarly, the consensus is lacking about relative contributions among causal determinants for species diversity[27–29] (L1, L3, and L6 in Fig. 1). Therefore, lack of proper quantification of individual causal links could fail to identify the most critical drivers in ecosystems and to reconstruct complex feedbacks under various environmental contexts.

Better quantification of more complex feedbacks (e.g., pairwise feedbacks and triangular feedbacks in Fig. 1b) is essential to predict responses of ecosystems to external perturbations[30]. Whereas major pairwise feedbacks indicate interactions in a network that potentially amplify or dampen external perturbations, quantifying triangular feedbacks can provide a better mechanistic understanding of these feedbacks and enable more accurate predictions for how changes in one variable will propagate to other parts of the network, producing more comprehensive, nonadditive impacts on ecosystems than biodiversity effects alone[8,31]. Thus, quantifying these complex network modules may have important management implications, in shifting the focus from managing individual state variables toward managing integrated ecosystem processes[32].

In this study, we used a combination of nonlinear time series methods, convergent cross-mapping (CCM)[33], and cross-system network analysis to elucidate the role of diversity and ecosystem functioning in natural aquatic ecosystems. CCM is a method rooted in the theory of dynamical systems[34] that enables the detection of causation between time series variables[33]. It is noteworthy that our current knowledge on interactions among biodiversity, ecosystem functioning, and environmental contexts is mainly derived from terrestrial ecosystems rather than natural aquatic ecosystems[12,35], despite phytoplankton accounting for >50% of global primary production[36]. Although effects of phytoplankton species diversity on biomass and resource use efficiency have been examined[35,37], the importance of diversity effects and more complex interaction modules remain unclear in natural aquatic systems. Therefore, we employed CCM[33] to assemble causal networks for 19 sites (Supplementary Fig. S1) among 16 globally distributed ecosystems ("Methods" and Supplementary Fig. S1), representing various types of aquatic ecosystems with various morphometrics and trophic states (from oligotrophic to eutrophic systems presented in Supplementary Table S1). Our datasets consisted of long-term monthly measurements (16–41 years) of phytoplankton data, with biodiversity and ecosystem function operationalized as phytoplankton species richness and community biomass (using chlorophyll-*a*

concentration as a proxy)[9], respectively. In addition, environmental variables, including concentrations of nitrate ($NO_3$) and phosphate ($PO_4$) (endogenous factors), and water temperature (an exogenous factor) were also involved in the reconstruction of causal networks for each system (see Supplementary Fig. S2 for an example, and similarly such a causal network comprised of the same variables as Supplementary Fig. S2 was reconstructed for each of the 19 sites). Based on the reconstructed networks, we aimed to understand biodiversity in aquatic ecosystems by addressing the following questions:

1. Under what conditions are phytoplankton diversity effects on ecosystem functioning stronger than the effects of environmental drivers?
2. What is the strongest causal determinant for species diversity?
3. What are the most effective pathways through which changes in diversity propagate to other parts of the network, and feedback on themselves?
4. Are there any emerging macroecological patterns explaining how causal links, pairwise feedbacks, and triangular feedbacks vary along large-scale environmental gradients?

To explicitly answer these questions, we performed cross-system comparisons on the reconstructed causal networks to evaluate: (i) the relative importance among causal links affecting phytoplankton biomass; (ii) the relative importance among causal links affecting phytoplankton diversity; (iii) the relative strengths of more complex feedbacks involving biodiversity; and iv) how the strengths of the network modules investigated in (i)–(iii) vary with environmental characteristics.

Overall, our analysis presents quantitative causal networks consisting of causal interactions and feedbacks among phytoplankton diversity, biomass, and environmental drivers and reveals how the network varies along large-scale environmental gradients. Our results indicate that phytoplankton diversity is a more important determinant to phytoplankton biomass than other environmental factors (nutrients and temperature) in more diverse, oligotrophic ecosystems; nutrients play an important role in determining the dynamics of phytoplankton diversity in most systems. In addition, strong nitrate–diversity–biomass feedbacks prevail in warm, productive systems; while strong phosphate–diversity–biomass feedbacks prevail in cold, less productive systems. These findings anticipate the response of aquatic ecosystems to environmental changes from a holistic network view.

## Results and discussion

**Quantification of causal networks.** We first compared the relative strengths of causal links across systems (Supplementary Fig. S3). Phytoplankton species richness was the major controlling factor for phytoplankton biomass (significant in 16 of 19 sites, Fig. 2a) in these diverse aquatic systems, consistent with experimental studies[17]. However, the averaged linkage strength for this effect was not significantly different from that of $NO_3$ (i.e., BD → EF vs. $NO_3$ → EF; permutation test $P = 0.501$), highlighting that nitrogen availability was equally important in affecting phytoplankton biomass in natural systems.

In the opposite direction, phytoplankton biomass was a significant driver of phytoplankton species richness in most ecosystems (15 of 19 sites, Fig. 2b). However, $NO_3$ more often had a stronger effect, appearing as the most important driver in 11 of 19 sites compared to phytoplankton biomass (4 of 19 sites) (Fig. 2b). Although the difference in effect strength was not significant (permutation test, $P = 0.162$), these results implicated nitrogen availability as an essential determinant affecting both phytoplankton diversity and biomass. As a sensitivity test, we also

examined the effects of Shannon diversity. The results suggest that the importance of nutrients is robust to the use of other diversity indexes (e.g., Shannon diversity in Supplementary Fig. S4), although the causal effects from phytoplankton biomass became relatively more important compared to biomass effects on species richness (Fig. 2b). Based on these findings, we inferred that processes influencing nutrients (e.g., external loadings and internal cycling[38]) need to be considered when investigating aquatic biodiversity. Changes in those processes (e.g., climatic[39] or anthropogenic[40] driven nutrient changes) may indeed substantially impact phytoplankton biodiversity, and subsequent ecosystem functioning.

The importance of $NO_3$ uncovered in our analyses might not be a counter-intuitive result, as many systems analyzed in this study were P-rich. For instance, the average phosphate concentration was 57.5 and 41.7 µgP/L for Lake Mendota (Me) and Lake Monona (Mo) (Supplementary Table S1), respectively. In addition, there were also high total phosphorus (TP) concentrations in shallow lake systems, e.g., average TP was 106.1, 112.5, and 126.4 µgP/L in Lake Inba (Ib), Lake Kasumigaura (Ks), and Müggelsee (Mu), respectively. Phosphorus was not always a limiting factor in eutrophic and mesotrophic systems, e.g., Lake Kasumigaura[41] and Lake Geneva (Gv)[42]. In addition, nitrogen was deficient and limited cyanobacteria bloom in Müggelsee (Mu)[43]. Nonetheless, we cannot exclude the possibility of colimitation[44] in N and P and the possibility that P availability also depends on N[45], which warrants further investigation.

Apart from nutrients and temperature, the causal effects of other important drivers on phytoplankton biomass and diversity were also examined, though not in all 19 systems due to data limitation. The causal effects of physical environmental factors, such as irradiance and water column stability, were presented in Supplementary Fig. S5; the results indicated that the quantified causal strengths on average were not as strong as the effects of diversity and nutrients. Moreover, the effects of consumers (e.g., zooplankton), which have been suggested as important drivers affecting species diversity of phytoplankton communities[46], were also examined. Based on our analysis of zooplankton, the causal effects of herbivorous crustaceans on phytoplankton biomass and diversity were significant in most of the analyzed systems. However, these effects were on average not as strong as the effects of phytoplankton diversity and nutrients, respectively (Supplementary Fig. S6). Nonetheless, these findings were not generalized to all 19 systems due to a lack of complete datasets as shown in Supplementary Table S3, and thus warrant more detailed investigation in future studies.

In addition to individual causal effects, we investigated feedbacks across systems. Pairwise feedbacks (e.g., BD ↔ EF and $NO_3$ ↔ EF) were common (Fig. 2c). However, the averaged linkage strength was often stronger in one direction when involving BD (Fig. 3). Specifically, the average strength of BD → EF was stronger than for the opposite direction of EF → BD (permutation test $P = 0.015$); BD → EF was stronger than EF → BD in 14 of the 19 systems (Fig. 3). In addition, biodiversity effects on nutrients (BD → $NO_3$ and BD → $PO_4$) were also stronger than their reversed effects ($NO_3$ → BD and $PO_4$ → BD) in 12 and 13 systems, respectively. In comparison, the interactions between nutrients and productivity were more symmetrical: nutrient effects on biomass ($NO_3$ → EF and $PO_4$ → EF) were stronger than biomass effects on nutrients (EF → $NO_3$ and EF → $PO_4$) in only 9 and 8 of 19 systems, respectively. These results supported the previous findings[8] that biodiversity effects more often operate at short-term scales, which makes effects more observable in our monthly-scale analyses than feedback effects on diversity, which are expected to occur on a more prolonged timescale, e.g., through slowly changing nutrient

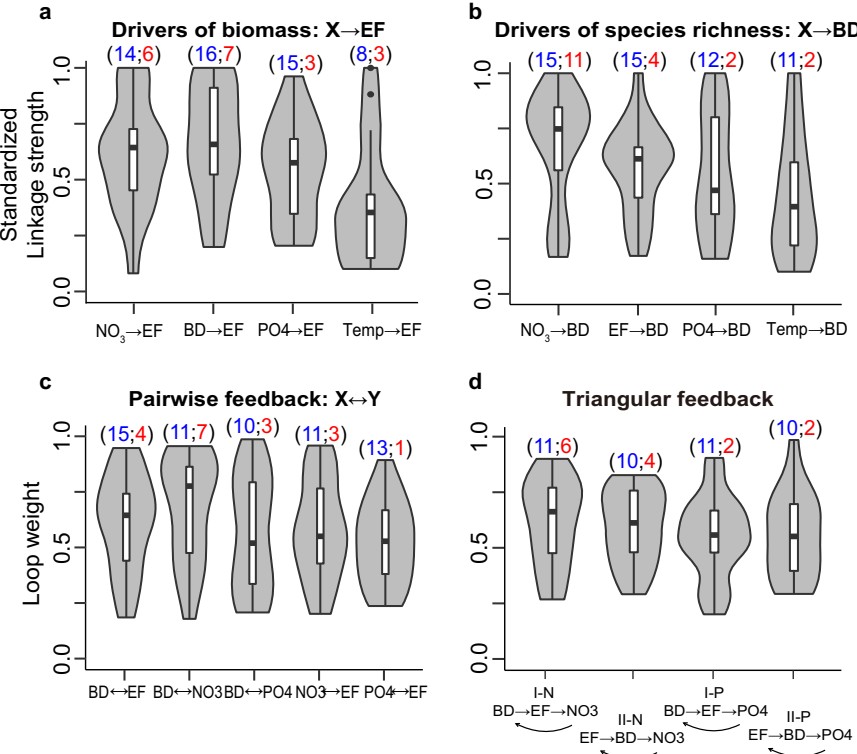

**Fig. 2 Relative strengths of various modules.** Standardized linkage strengths of causal variables affecting (**a**) phytoplankton biomass and (**b**) species richness (here, BD) and loop weights for various types of (**c**) pairwise feedbacks and (**d**) triangular feedbacks. All statistics were calculated from the 19 independent sites ($n = 19$) and depicted as joint violins and box plots to present the empirical distribution that labels the maxima and minima at the top and bottom of the violins, respectively, and shows 25, 50, and 75% quantiles in the boxes with whiskers presenting at most 1.5 * interquartile range. The two numbers within the parentheses ($S$; $R_1$) above each violin plot report the number of significant results in CCM ($S$; labeled blue) and the number of systems in which a particular module had the greatest strength (i.e., rank 1; R1; labeled red). Source data are provided as a Source Data file.

cycling[31] or decomposition[47]. Nevertheless, the timescale dependence of causal interactions in ecosystem networks is a topic that needs further study.

Subsequently, we quantified the strengths of pairwise feedbacks as the geometric mean of the linkage strengths in each direction, following a previous study[9] (see more details in Methods). Among these feedbacks (Fig. 2c and Supplementary Fig. S7), BD ↔ NO₃ had the highest median and average strength (0.78 and 0.68, respectively) across systems. However, strengths of BD ↔ NO₃ were highly variable among systems (large interquartile range in Fig. 2c), and thus were only significant in 11 of 19 systems, compared to BD ↔ EF (15 of 19 systems). These findings reinforced the importance of nutrients as key determinants for aquatic biodiversity and implied that nutrient effects are context-dependent. In other words, BD ↔ NO₃ was less common than BD ↔ EF across systems, despite its stronger average strength. The prevalence of BD ↔ EF indicated a need for more long-term experiments and process-based/theoretical modeling accounting for *bidirectional* interactions between diversity and biomass[16], because bidirectional interactions and feedbacks may challenge our simple predictions for ecosystem dynamics, based on knowledge of unidirectional interactions[30].

Quantification of the causal network also allowed us to analyze triangular feedbacks. Within the conceptual framework of Fig. 1b, there are four kinds of triangular feedbacks involving biodiversity, ecosystem functioning, and either nitrate or phosphate (Type I: BD → EF → NO₃ and BD → EF → PO₄; Type II: EF → BD → NO₃ and EF → BD → PO₄). There was at least one significant triangular feedback in 14 of 19 sites (Fig. 2d). More specifically, NO₃-associated feedbacks (Type I-N and Type II-N) were usually stronger than PO₄-associated feedbacks (Type I-P and Type II-P)

(Fig. 2d), although the difference in strength among the four types of feedbacks was not significant (Fig. 2d; Kruskal–Wallis test, $P = 0.59$). The dominance of NO₃-associated feedbacks in our study was attributed to many of the sites being marine and eutrophic lakes, which are likely to be N-limited due to an imbalance in external loadings[48] or strong denitrification[49]. Among both NO₃- and PO₄-associated feedbacks, there were no significant differences in strength between Type I and Type II feedbacks (Supplementary Fig. S7), suggesting that biodiversity can directly influence biomass (Type I), as well as through a pathway that involves endogenous nutrient variables (Type II) and eventually feeds back on itself.

**Causal networks under environmental contexts.** Our empirical analyses revealed state dependency of the causal links and feedbacks among biodiversity, biomass, and environmental factors in natural systems; that is, their strengths were highly dependent on the state of other variables. Based on a cross-system comparison (Methods), strengths of individual links (e.g., BD → EF), pairwise feedbacks (e.g., BD ↔ EF), and triangular feedbacks (e.g., BD → EF → NO₃ → BD) varied systematically, depending on environmental characteristics (Fig. 4 and Supplementary Fig. S8). Ecosystems with higher species diversity (long-term average species richness) and lower average PO₄ concentrations had stronger BD → EF links (Fig. 4a; correlation coefficient $r = 0.600$ and $-0.513$; $P = 0.007$ and 0.025 for species diversity and PO₄, respectively). These results were further confirmed by stepwise regression, indicating that the ecosystems characterized by higher diversity, lower average temperature, and oligotrophic conditions had stronger BD →

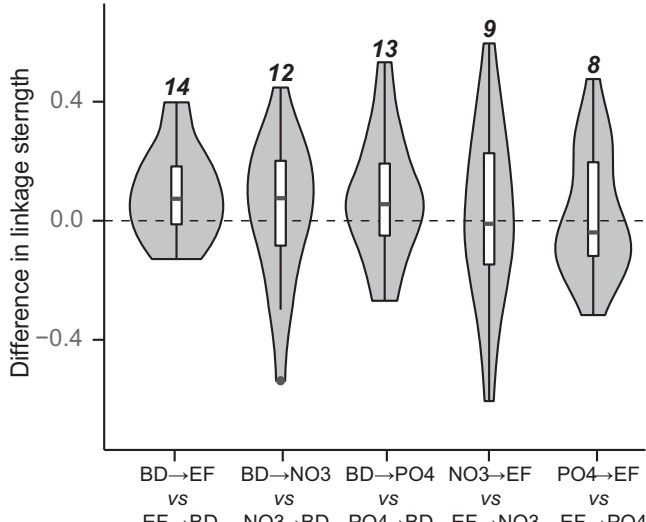

**Fig. 3 Directional bias in pairwise feedbacks.** The difference in standardized linkage strengths between the two directions was computed for each pairwise feedback and depicted as joint violin and box plots. All statistics were calculated from the 19 independent sites ($n = 19$) and depicted as joint violins and box plots to present the empirical distribution that labels the maxima and minima at the top and bottom of the violins, respectively, and shows 25, 50, and 75% quantiles in the boxes with whiskers presenting at most 1.5 * interquartile range. The number above the plot indicates the number of systems with a positive difference in linkage strength. For example, BD → EF was stronger than its feedback, EF → BD, in 14 of the systems. In general, the strength of diversity effects (BD → EF, BD → $NO_3$, BD → $PO_4$) was usually stronger than feedback effects (EF → BD, $NO_3$ → BD, $PO_4$ → BD). Source data are provided as a Source Data file.

EF (best-fit regression model: BD → EF strength = $0.663 + 0.171*BD - 0.139*T - 0.096*PO_4$; $F_{3, 15} = 9.958$ and $P < 0.001$). In contrast, temperature and $PO_4$ effects on phytoplankton biomass (i.e., T → EF and $PO_4$ → EF) were negatively associated with long-term average species diversity, but positively associated with average $PO_4$ (Fig. 4a). Therefore, we inferred that phytoplankton biomass in P-rich systems was more sensitive to warming (due to strong T → EF). This synergistic effect of warming and eutrophication on biomass has been reported in other aquatic ecosystems[50]; in this study, this synergistic effect was weaker when species diversity was higher and BD → EF was stronger (Fig. 4a). Perhaps greater diversity and its effects mitigate adverse impacts of global warming[9], although warming may also weaken biodiversity effects on ecosystem functioning due to strong interspecific competitions under high temperatures[51].

In the opposite direction, stronger EF → BD was associated with lower temperature and $PO_4$ concentrations, as well as shallower depths (Fig. 4b; best-fit regression model: EF → BD strength = $0.572 - 0.170*PO_4 - 0.101*Depth - 0.096*T$; $F_{3, 15} = 9.800$ and $P < 0.001$). In shallower systems, which are better mixed and less vertically heterogeneous, impacts of species competition on diversity may be more influential[52]. In contrast, the effects of the most important driver on diversity, $NO_3$ → BD, exhibited an opposite response to temperature (Fig. 4b), suggesting that the dominant determinants of aquatic biodiversity varied along a temperature gradient.

Water temperature and phytoplankton biomass (Chl*a* as a proxy) also critically determine strengths of various pairwise feedbacks. Stronger $PO_4$-mediated feedbacks (BD ↔ $PO_4$ and EF ↔ $PO_4$ in Fig. 4c) were usually more associated with cold

($r = -0.247$ and $-0.329$, respectively) and less productive systems ($r = -0.421$ and $-0.527$, respectively); this contrasted with BD ↔ $NO_3$, which was more associated with warm (Fig. 4c; $r = 0.503$) and productive environments ($r = 0.571$). This finding was consistent with the notion that N is more often a limiting element for phytoplankton growth in warm, tropical/subtropical systems than in cold, temperate systems[53]. However, EF ↔ $PO_4$ and BD ↔ EF had no clear relationship with temperature ($r = 0.167$ and $-0.177$, respectively) or biomass ($r = 0.284$ and $0.163$, respectively). Thus, we speculated that climate warming will shift aquatic ecosystems towards stronger coupling in biodiversity-$NO_3$ feedback than biodiversity–biomass feedback.

Our analyses improved understanding of how more complex regulations varied with environmental characteristics. In our study, strengths of triangular feedbacks, regardless of direction, were positively associated with high diversity environments. Of the four triangular feedbacks, both type I-N feedbacks (BD → EF → $NO_3$) as well as the type II-N feedback (EF → BD → $NO_3$), were positively associated with high-diversity environments (Fig. 4d, $r = 0.531$ and $0.561$, respectively). Therefore, we inferred that the dynamics of biodiversity, biomass, and endogenous variables were tightly coupled in high-diversity systems; that is, dynamics of one component were more responsive to changes in other parts of the feedback. Our findings contrasted with the prevailing view that ecosystem functioning is insensitive to changes in diversity at high levels of diversity (i.e., the redundancy concept[18]), prompting a further investigation to clarify the role of biodiversity in regulating ecosystem dynamics[9].

Interestingly, triangular feedbacks in different directions (i.e., Type I versus II) had distinct responses to biomass levels. The strength of Type I feedbacks had no statistical association with biomass, implying that diversity effects on biomass (i.e., BD → EF) can propagate to nutrients (i.e., EF → nutrients) and then to diversity itself (i.e., nutrients → BD), irrespective of biomass levels. In contrast, Type II feedbacks had associations with biomass; the latter was positively associated with strengths of Type II-$NO_3$ feedbacks ($r = 0.567$; $P = 0.011$), but negatively associated with strengths of Type II-$PO_4$ feedbacks ($r = -0.430$; $P = 0.066$). This highlighted the importance of considering linkage directionality when studying these regulatory feedbacks under various environmental contexts. For example, even when the same components were considered, two causal interactions in the opposite direction (e.g., BD → EF or EF → BD) responded to environmental factors differently (e.g., Fig. 4a, b, respectively).

Our network-based approach enabled us to describe how responses of triangular feedbacks to environmental changes differed from responses of individual links or pairwise feedbacks. Different regulatory feedbacks were associated with distinctive environmental characteristics (Fig. 4d) and not necessarily similar to that of individual links involved. For instance, EF → BD-driven triangular feedbacks (Type II) were associated with average phytoplankton biomass (Fig. 4d), but phytoplankton biomass was associated with neither the EF → BD nor BD ↔ EF, individually. Indeed, the cross-system patterns of triangular feedbacks were statistically distinguishable from that of pairwise feedbacks (Supplementary Fig. S9), implying unique responses of complex feedbacks to environmental gradients. Therefore, predicting ecosystem responses to environmental changes is challenging, even if responses of individual links or of pairwise feedbacks can be elucidated, because quantification of individual causal links in isolation might fail to recover more complex network modules. As feedbacks and other network modules[54] are critical for stabilizing/destabilizing ecosystem dynamics[5,55], it is becoming apparent that studying interdependencies among key ecological processes from a holistic network view is needed for predicting ecosystem dynamics under environmental changes.

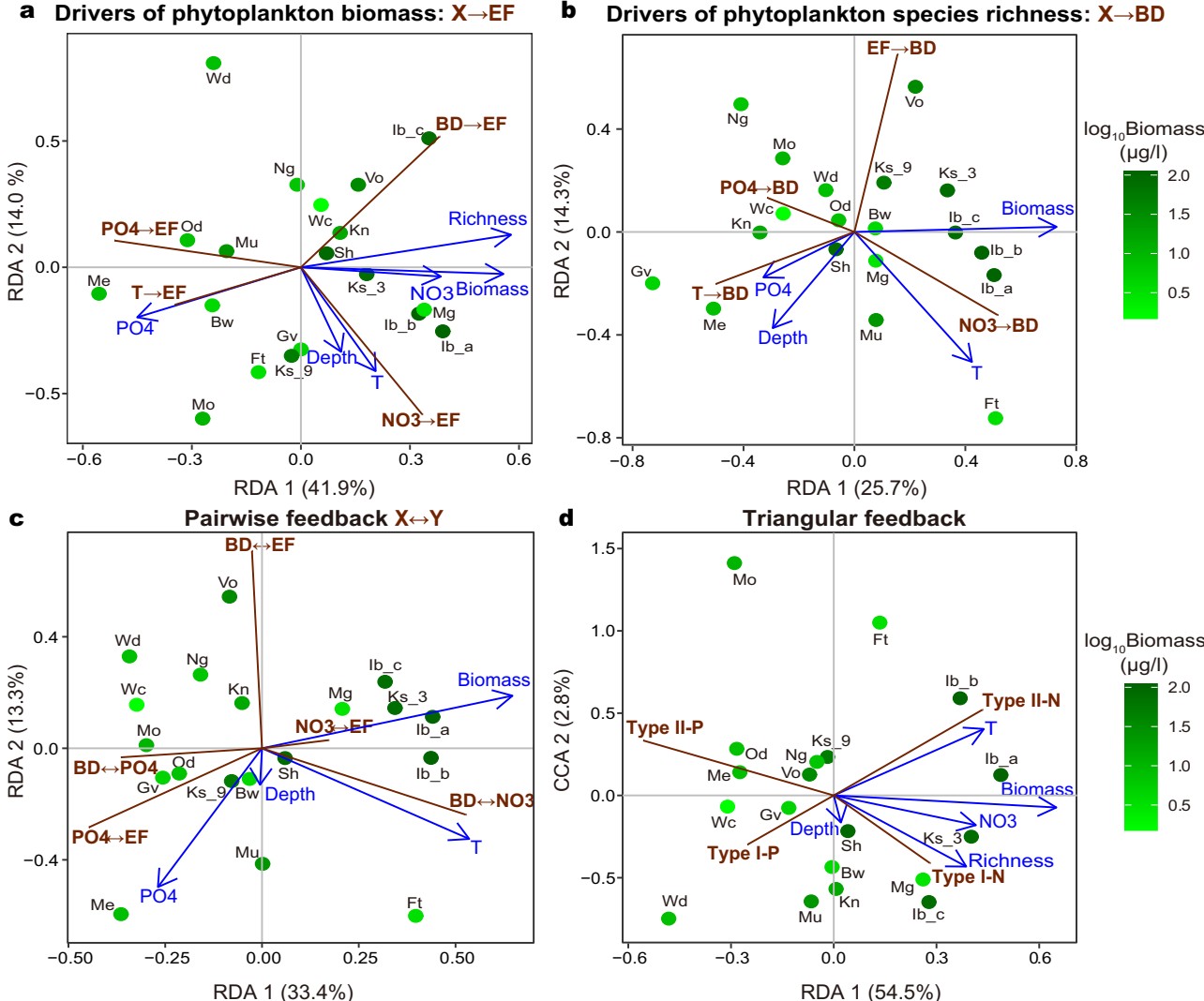

**Fig. 4 Strengths of causal interactions and feedbacks vary along environmental gradients of aquatic ecosystems.** Multivariate ordination illustrating associations between long-term averages of environmental factors (blue) and quantitative network modules, including the strength of links affecting: **a** phytoplankton biomass (X → EF; red) and (**b**) species richness (X → BD; red), and the loop weight of (**c**) pairwise feedbacks and (**d**) triangular feedbacks. Here, water temperature was abbreviated by T. Because biomass was selected throughout the analyses, we used color scales (green) to visualize data points according to their $\log_{10}$ biomass values. A similar figure based on the color scales of water temperature was presented in Supplementary Fig. S8. Black text indicates abbreviated site names (see Supplementary Fig. S1). **a** Stronger BD → EF was observed in environments characterized by low $PO_4$ but high diversity. In comparison, T → EF and $PO_4$ → EF were associated with high $PO_4$ but low diversity, and $NO_3$ → EF was associated with high temperature. **b** Stronger EF → BD was observed in the environments characterized by intermediate level of phytoplankton biomass. In comparison, stronger T → BD and $PO_4$ → BD were usually present in systems with a lower phytoplankton biomass. However, $NO_3$ → BD was associated with high-temperature environments. By including feedbacks of diversity and biomass on endogenous environments, we determined that: **c** stronger diversity–biomass feedbacks were more often observed in phosphate-poor systems. Diversity-nitrate feedbacks (BD ↔ $NO_3$) were stronger in high-temperature environments; diversity-phosphate (BD ↔ $PO_4$) feedback were weaker in low biomass environments. **d** Stronger phosphate-associated feedbacks (Type I-P and Type II-P) were more often observed in systems characterized by lower temperature, phytoplankton biomass, and nitrate ($NO_3$). In contrast, stronger nitrate-associated feedbacks (Type I-N and Type II-N) were more often observed in environments with higher temperature (especially for Type II-N), biomass, and species diversity. Selected environmental variables significantly explained the multivariate ordinations (permutation test $P = 0.004$, $0.029$, $<0.001$, and $0.001$ for Panels **a** to **d**, respectively). Source data are provided as a Source Data file.

**Sensitivity analysis using composition-converted biomass measure.** Our conclusions were robust to the use of alternative biomass measure. Specifically, the main findings (Figs. 2 and 4) based on the analysis of phytoplankton biomass inferred by Chl*a* were qualitatively similar with findings based on composition-converted biomass (Supplementary Figs. S10 and 11). Although the relationship between Chl*a* and true phytoplankton biomass varies with environmental conditions (e.g., light), it remains an effective functional index inferring phytoplankton stock with more emphasis on photosynthesis capacity (i.e., biomass of photosynthetic machines). In contrast, composition-converted biomass, though has similar meaning with overall phytoplankton biomass, contains high uncertainty by assuming species-specific conversion factors, especially when the measurement of individual cells size was lacking. In addition, these conversion factors were determined by various geometrical models, which differ among systems; this is in contrast to the standard chemical approach used in determining Chl*a*, which makes Chl*a* more

suitable to reveal cross-system variations in causal strengths along environmental gradients (Fig. 4 and Supplementary Fig. S11e–h). However, Chl*a* integrates all kinds of photo-autotrophs that might not be fully included in counting data (e.g., picoplankton). Thus, investigating the diversity effects of more complete phytoplankton groups requires novel techniques (e.g., metagenomics[56]). Nonetheless, it remains an open question about how the presented causal links and feedbacks change when considering various types of functional indices and diversity measures.

**Caveats for the reconstruction of causal networks in natural phytoplankton communities**. Several issues warrant further studies in aquatic ecosystem networks involving phytoplankton diversity and biomass. Firstly, the number of marine sites was limited, hindering comparisons of marine versus freshwater systems. Furthermore, our analyses based on CCM cannot access the sign of feedbacks (i.e., positive or negative), although it is known that the sign is important in determining the response of feedbacks to external perturbations (e.g., amplified or dampened). Although methods to estimate the sign of interactions were proposed (e.g., S-map[22,57]), the robustness of these methods has not been thoroughly examined[58]. Lastly, due to limitations of data availability, our analysis only quantified causal strength across systems at a consensus monthly scale, acknowledging that state-space reconstruction methods (e.g., CCM) are scale-dependent[59], e.g., one causal driver dominated monthly might not necessarily dominate at other time scales. Therefore, exploring causal feedbacks at other time scales needs further investigations by including more datasets with high temporal resolution and long-duration monitoring.

**Final remarks**. Our findings bridged two popular and contrasting research directions: whereas many studies consider diversity effects on ecosystem functioning[8,60], other studies aim to identify determinants of species diversity[61], which can be traced to Hutchinson's seminal question about species coexistence[62]. Our findings highlighted that these two ecological processes are interdependent and embedded in a complex network. Thus, these ecological processes should not be investigated in isolation[10,12,30], but instead be examined in integrated feedbacks, especially for rapid turnover systems (e.g., plankton or microbial communities in aquatic systems) in which feedbacks from biomass to diversity can operate quickly through light shading[29] or exploitation on nutrients[63]. It is noteworthy that our proposed methodological framework can be applied to explore in more detail causal feedbacks or paths if precise mechanistic measures (e.g., nutrient recycling rate rather than nutrient stock) can be monitored over time. For instance, biodiversity was suggested to influence ecosystem functioning via species complementarity or selection effects. Measuring complementarity or selection effects is available for experimental data[64], but remains a challenging task for observational data. Thus, the incorporation of these detailed mechanistic measures in the causal networks is an important future research topic. More comprehensive surveys are required in future ecological monitoring to improve our understanding of causal mechanisms embedded in causal networks.

Our analyses quantified diversity-associated causal networks among various natural aquatic ecosystems. The selected long-term datasets from various aquatic ecosystems represented a reasonable parallel to long-term biodiversity experiments conducted in terrestrial grassland ecosystems[65]. Revealing the dominance of diversity effects on biomass in these systems (Fig. 2) was enabled by using methods for nonlinear dynamical systems, in lieu of linear statistical analyses[19,25]. Indeed, when

linear analyses were applied in our datasets (Supplementary Fig. S12), the importance of diversity effects on biomass (BD → EF) and nitrate effects on diversity (NO₃ → BD) were not clearly identified as that shown in the nonlinear CCM analysis (Fig. 2a, b).

Through cross-system comparison, the strength of causal relationships associated with species diversity varied with nutrient levels and time-averaged mean levels of diversity in aquatic systems (Fig. 4a). Associations with environmental gradients were also present in other network modules (i.e., feedbacks; Fig. 4c, d). These statistical associations revealed the macroecological relationships of how the strength of biodiversity effects and related feedbacks varied with environmental gradients. Moreover, the unveiled macroecological patterns also improved our understanding of how causes and effects of biodiversity are modulated by biotic and abiotic contexts. Although the proposed relationships need to be examined through more long-term experiments, our quantitative and empirical framework for constructing causal networks provided a foundation for better predicting the consequences of biodiversity loss across ecosystems.

## Methods

**Data**. Phytoplankton species composition and environmental data were compiled from 16 aquatic ecosystems with a total of 19 long-term monitoring sites, spanning a large range of freshwater and marine types, from shallow to deep and from oligotrophic to eutrophic, as follows (Supplementary Table S1 and Supplementary Fig. S1): (1) Lake Biwa, Japan, 1978–2010; (2) Feitsui Reservoir, Taiwan, 1986–2017; (3) Lake Geneva, France/Switzerland, 1974–2014; (4) Lake Inba, Japan, 1986–2016 (including three stations in disparate lake basins); (5) Lake Kasumi-gaura, Japan, 1978–2009 (including two stations in distinct lake basins); (6) Lake Kinneret, Israel, 1996–2012; (7) Lake Maggiore, Italy, 1997–2015; (8) Lake Mendota, USA, 1995–2012; (9) Lake Monona, USA, 1995–2011; (10) Müggelsee, Germany, 1994–2013; (11) Narragansett Bay, USA, 1999–2014; (12) Lake Oneida, USA, 1975–1995; (13) Shin River, Japan, 1986–2016; (14) Lake Võrtsjärv, Estonia, 2001–2016; (15) Station L4, Western English Channel, England, 1992–2009; and (16) Windermere, England, 1993–2010 (South basin).

For all systems, there were five types of variables: (1) phytoplankton species richness (number of species recorded in a sample); (2) chlorophyll-*a* concentration as a measure of phytoplankton biomass and ecosystem function, a widely used proxy of algal biomass in the BDEF literature[17,66]; (3) phosphate concentration (PO₄); (4) nitrate concentration (NO₃); and (5) water temperature. Phytoplankton samples were identified to the finest taxonomical level (generally species level if possible) and enumerated under an optical microscope, based on counting methods summarized in Supplementary Table S2. The counting methods used were similar (e.g., Utermöhl[67] method and relevant approaches). Based on composition data, species richness was derived and defined as the number of species present in the phytoplankton community. In systems with depth-resolved measurements, data were depth-integrated averages in the euphotic zone; otherwise, measurements were from surface layer samples.

We additionally examined other environmental factors that are known to be important to phytoplankton, including water column stability, irradiance, and zooplankton abundance, although these variables were not measured in all systems (Supplementary Table S3). Water column stability was calculated as maximal Brunt-Väisälä frequency from temperature vertical profile data as an index of water column stability; irradiance data was compiled from in situ measurements of weather or buoy stations near the sampling sites. For zooplankton analysis, we compiled composition and density data (individual/L) of crustacean zooplankton in 11 sites (Supplementary Table S3) based on microscopic counting. Specifically, we investigated grazing effects of the following three zooplankton categories: (i) herbivorous cladocerans excluding predatory taxa, such as *Bythotrephes* spp. and *Leptodora* spp.; (ii) herbivorous copepods including all calanoids and naupliar stages of cyclopoids; and (iii) herbivorous crustaceans including both herbivorous cladocerans and herbivorous copepods (i.e., i+ii). It is noteworthy that our zooplankton analysis was based on density instead of biomass data because zooplankton length measurements, which are required for converting individual counts to biomass data, were absent in 5 of the 11 analyzed sites.

**Data treatment**. For consistency, monthly time series were generated by averaging over observations if sampling occurred on a finer timescale. Although such compilation potentially causes some inconsistency in smoothing temporal fluctuations of time series data among systems with various sampling frequencies, it was necessary because our methods based on state-space reconstruction require time series data at equal intervals, dictating the temporal scale of analysis. In our case, the monthly resolution is the only consensus that can be applied to all time series datasets and the monthly average is the most representative measure at this scale.

Nonetheless, causal strengths estimated by CCM analysis were robust to this data averaging according to our comparisons using eight stations where regular and frequent sampling (i.e., sampling frequency higher than monthly) were available (Supplementary Fig. S13). Overall, our data compilation yielded 5554 data points for each variable across the 19 sites (Supplementary Table S1). To ensure stationarity, we removed the long-term linear trend from each time series by using the residuals from a linear regression against time[9]. We accounted for seasonality by scaling against the mean and standard deviation of values occurring in the same month[9], $D_{-mv}(t_i) = (O(t_i) - \mu_{month\ i})/\sigma_{month\ i}$, where $\mu_{month\ i}$ is the monthly mean, $\sigma_{month\ i}$ is the monthly standard deviation for each of 12 months, $O(t_i)$ is the original time series, and $D_{-mv}(t_i)$ is the deseasonalized time series, $i = 1, 2, \ldots, 12$. Finally, each time series was re-scaled to zero long-term mean and unit variance[68].

**Convergent cross-mapping analysis.** Causal networks were reconstructed among phytoplankton species richness, phytoplankton biomass, and the environment with a method specifically designed for quantifying causality in nonlinear dynamical ecosystems, convergent cross-mapping (CCM)[33]. In that regard, CCM is a causality analysis based on Takens' theorem for dynamical systems[34,69], which infers the causal relationship among variables from their empirical time series. CCM[33] tests for causality between pairs of time series by measuring the extent to which the historical record of an effect variable, $X$, can reliably estimate states of a causal variable, $Y$[33,69,70]. Cross-map skill, the quantification of this measure, is defined as the correlation coefficient $\rho$ between estimated states of the causal variable and actual observations[71]. CCM is based on information recovery (i.e., effect variables contain encoded information on causal variables), instead of predictive ability (using causal variables to predict future values of effect variables, e.g., Granger's causality). The essential ideas of CCM are summarized in the following brief animations: tinyurl.com/EDM-intro. The aforementioned deseasoning procedure reduces detection of false positives caused by 'dynamical synchronization'[72,73] under strong seasonality. Apart from seasonality, dynamical synchronization can also occur when interactions between two variables are very strong[33]; nevertheless, very strong interactions are of less concern here because most interactions in real ecosystems are weak to moderate[74]. A modeling study also indicated that CCM was robust against moderate noise from process and observational errors[75].

Several limitations in applying CCM analysis need to be acknowledged. First, CCM is based on lagged coordinated embedding in which each embedded variable needs to be lagged by a fixed time interval that determines the timescale of CCM analysis. For example, time series analyzed in our study were integrated to the monthly scale as stated in "Data treatment" section. Multi-scale analysis is possible only when data points are measured very regularly across various time scales (e.g., from weekly to monthly). Second, as required in many time series analysis, CCM analysis requires time series data being stationary[76]. Otherwise, CCM likely produces false-positive findings (e.g., caused by strong seasonality); note however, we have removed seasonality in analysis in this work. Third, a time series including too many zero values (or other constant values) is not suitable for CCM analysis (as a general statistical issue in any time series analysis). This is because embedding such a time series potentially produces many zero vectors, which violates the general assumption of EDM that assumes a one-to-one mapping between each embedded vector and the vector on dynamical manifold[34] (i.e., zero vector can map to many possibilities on manifold). Thus, the embedded zero vectors need to be excluded or separated from the prediction set[73].

CCM analysis accounts for influences of confounding variables implicitly. Specifically, CCM incorporates influences of confounding variables using lagged embeddings, e.g., $(X_{t-1}, X_{t-2}, \ldots)$, which have accounted for historical effects of other variables in lagged terms, even if those variables were unobserved or difficult to identify. As such, CCM does not require identifying or ruling out influences of confounding variables in order to quantify causations between two variables, and thus can be applied in more general dynamical systems[68]. In addition, CCM is a nonparametric approach, free from assumptions of the specific form of quantitative relationships between causal variables. Although this makes CCM difficult to explore quantitative features, e.g., the minimal number of species required to maintain 80% levels of ecosystem function, it provides high flexibility to infer causations in nonlinear dynamical systems. Such flexibility is important for inferring nonlinear dynamical systems, because quantitative relationships between any two dynamical variables could change, depending on the varying state of other state variables[77] or environmental contexts. For example, linear associations between two variables will appear then disappear or change sign—so-called mirage correlations[33], making methods based on modeling static, parametric relationships difficult to correctly identify causations[9].

In this study, the best embedding dimension ($E$) used in CCM was determined by testing a range from 2 to 20 and selecting the value that optimized the hindcast cross-mapping in which $X(t)$ projected one-step backward to $Y(t-1)$; this avoids overfitting of $E$ when $X$ and $Y$ are unrelated time series[73]. Note that $E$ can vary for each variable pair: the $E$ selected for $X$ cross-mapping $Y$ can be different from the $E$ selected for $Y$ cross-mapping $X$ or $X$ cross-mapping $Z$.

The possibility of lagged causal effects[78] was explored in this study. Specifically, we tested for the causal influence of $Y$ on $X$ by cross-mapping between $X(t + k)$ and $Y(t)$ using time lags ($k$) of 0, 1, 2, or 3 months (corresponding to the timescale of phytoplankton dynamics) and selecting the mapping with the highest cross-map skill. To determine the convergence in cross-mapping, we followed the procedure in Sugihara et al.[33] and computed the cross-map skill for subsamples of $X(t)$ with varying library lengths ($L$). Here, the minimal library length, $L_0$, is equal to the embedding dimension, and the maximal library length, $L_{max}$, is equal to the length of the whole time series. To test the convergence of CCM, we applied two statistical criteria. First, we tested whether there was a significant monotonic increasing trend in cross-map skill, $\rho(L)$, using Kendall's $\tau$ test. Next, we tested the significance of improvement in cross-map skill using Fisher's $\Delta\rho$ $Z$ test and compared the cross-map skill for the maximal library size ($\rho(L_{max})$) against the cross-map skill at the minimal library length ($\rho(L_0)$).

**Quantification of causal interaction and loop weight.** As in previous studies[75,76], the strength of causal interaction was quantified based on the cross-mapping skill at the maximal library length, $\rho(L_{max})$. That is, stronger causal effects result in convergence to high cross-map skill[9,33,76]. For instance, a strong causal effect of species richness on phytoplankton biomass revealed by CCM indicated that dynamics of phytoplankton biomass (magnitude or variability) responded strongly to changing species richness. To address systematic differences in cross-map skill among study sites, which may arise due to differences in noise or time series length, causal strength[9] was standardized. The standardized linkage strength (SLS) was calculated by dividing linkage strength (LS) by the maximum within each system: SLS = LS/max(LS); thus, SLS varied between 0 and 1 and indicated the relative importance with respect to the strongest causal link within the system. It is noteworthy that causal networks were constructed and standardized separately for each system; i.e., it was not assumed that each system had equivalent dynamics and belonged to the same attractor.

Pairwise and triangular feedbacks were quantified using Neutel's loop weight[55], the geometric mean of SLS for all links within a given feedback. In a pairwise feedback (X ↔ Y), the loop weight is the geometric mean of the SLS in both directions (i.e., X → Y and Y → X). We classified two types of triangular feedbacks, based on the directionality of the involved interactions: "Type I", richness→biomass→nutrients→richness and "Type II", biomass→richness→nutrients→biomass. The Type I feedbacks occur in the direction that includes biodiversity effects on ecosystem function (BD → EF), whereas Type II feedbacks are in the opposite direction and include EF → BD. With two types of nutrients stocks (phosphorus P or nitrogen N), there are a total of four triangular feedbacks (I-N, I-P, II-N, and II-P). To determine the uncertainty of our estimates in causal strength and loop weight, we calculated their standard errors using resampling method that reconstructed sampling distributions from 500 random samples of embedded data points with replacement.

**Linking strengths of links and feedbacks with ecosystem characteristics.** Multivariate redundancy analysis (RDA)[79] was used to illustrate how the strength of individual links and feedbacks varied in association with ecosystem characteristics: depth and area of the study site, as well as long-term averages of species richness, temperature, phosphate, and nitrate. Specifically, we conducted RDA analyses for: (i) causal effects on phytoplankton biomass (Fig. 4a); (ii) causal effects on species richness (Fig. 4b); and (iii) pairwise and triangular feedbacks (Fig. 4c, d), respectively. For each multivariate RDA ordination, we constructed the biplot using the first two RDA scores to demonstrate how the prevalence of various links or feedbacks were statistically associated with various environmental characteristics. The use of RDA instead of CCA was justified based on our analysis on coenoclines (Supplementary Fig. S14), with more linear coenoclines and a short gradient length (<3)[80]. RDA significance was evaluated using a permutation test[79]. All permutation tests performed in this study were based on the null distribution generated from 10,000 random permutations. To further support the RDA results, correlation analysis was performed between the strength of key network modules and environmental characteristics and tested significance using a permutation test. These analyses were not intended to examine causation between long-term environmental characteristics and strength of network modules, but rather to provide a picture describing under what environmental conditions a module of interest (e.g., BD → EF) prevailed.

**Computation.** All analyses were done with R (ver. 4.0.3). The CCM analyses and the multivariate RDA analysis were implemented using the rEDM[81] and vegan[82] packages, respectively.

**Reporting summary.** Further information on research design is available in the Nature Research Reporting Summary linked to this article.

## Data availability
Raw time series datasets from all research sites are available, on request, through the paths listed in Supplementary Table S2 due to various data use policy. Source data for all figures are provided with this paper and available from Github online repository, https://github.com/biozoo/Chang_etal_2022_SI_CausalFeedback[83]. Source data are provided with this paper.

## Code availability

Documentation of all analytical procedures provided as R codes are available from Github, https://github.com/biozoo/Chang_etal_2022_SI_CausalFeedback[83].

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

## Acknowledgements

This work was supported by the National Center for Theoretical Sciences, National Taiwan University, Academia Sinica, Foundation for the Advancement of Outstanding Scholarship, and the Ministry of Science and Technology, Taiwan (to C.H.H.). Data for Oneida Lake were collected with support from Cornell University's Brown Endowment, New York State Department of Environmental Conservation, and United States Department of Agriculture, National Institute of Food and Agriculture, Hatch Project 0226747. Research on Lake Maggiore is within the framework of the LTER Italian and European networks, site 'IT-08 Southern Alpine lakes" and funded by the International Commission for the Protection of Swiss-Italian Waters (CIPAIS). Data for Lake Geneva were contributed by the Observatory of alpine LAkes (OLA), © SOERE OLA-IS, AnaEE-France, INRAE of Thonon-les-Bains, CIPEL. Data for Lake Võrtsjärv were provided by Estonian Environment Agency and by the Centre for Limnology at Estonian University of Life Sciences funded by the Estonian Research Council grants PRG 1167 and PRG709. This project has received funding from the European Union's Horizon 2020 research and innovation programme under grant agreement No 951963. Data for Lake Kinneret were collected by the Kinneret Limnological Laboratory, Israel Oceanographic & Limnological Research, and funded by the Israel Water Authority, with phytoplankton counts made by T. Fishbein, chlorophyll determined by Y. Yacobi, physical data collected by Y. Lechinsky, chemical analyses conducted by Mekorot Water Company, Watershed Unit, under oversight by A. Nishri, and database management services provided by M. Shlichter. Data from Lake Inba were provided by Chiba Prefecture. Data from Lake Kasumigaura were provided by the National Institute for Environmental Studies (NIES). Data from Lake Biwa were provided by Shiga Prefecture. Data for Müggelsee were provided by the Leibniz-Institute of Freshwater Ecology and Inland Fisheries within their long-term research programme. Data collection at Windermere was supported by Natural Environment Research Council award number NE/R016429/1 as part of the UK-SCaPE program delivering National Capability. Data for Station L4, Western Channel Observatory were collected by Plymouth Marine Laboratory as part of the UK's Natural Environment Research Council's National Capability CLASS Programme grant number NE/R015953/1, and is a contribution to Theme1.3 - Biological Dynamics. This is a contribution of GEISHA project, which was jointly supported by the French Foundation for Research on Biodiversity (FRB) through its synthesis center CESAB (http://www.cesab.org/) and the John Wesley Powell Center for Analysis and Synthesis (https://powellcenter.usgs.gov/). Comments from B. Kraemer, M. Kondoh, M. Ushio, and P. J. Ke on an earlier draft and English editing by J. Kastelic improved the manuscript.

## Author contributions

C.W.C., C.H.H., and T.M. conceived the research idea. C.W.C. analyzed the data with help from C.H.H., H.Y., and S.S. R.A., O.A., S.J.T., Y.B., Y.R.C., H.F., S.I., Ma.K., Mi.K., S.I.M., P.N., M.R., F.K.S., C.E.W., J.T.W., T.Z., H.A., G.G., S.B., X.L., M.M.M., and R.P. collected the data. C.W.C., C.H.H., T.M., and H.Y. wrote the draft of the manuscript with critical comments from all co-authors.

## Competing interests

The authors declare no competing interests.

## Additional information

[1]National Center for Theoretical Sciences, Taipei 10617, Taiwan. [2]Research Center for Environmental Changes, Academia Sinica, Taipei 11529, Taiwan. [3]Faculty of Advanced Science and Technology, Ryukoku University, Otsu, Shiga 520-2194, Japan. [4]Institute of Oceanography, National Taiwan University, Taipei 10617, Taiwan. [5]Center for Biodiversity Science, Ryukoku University, Otsu, Shiga 520-2194, Japan. [6]Health Science

Center Libraries, University of Florida, Gainesville, FL 32611, USA. [7]Univ. Lille, CNRS, Univ, Littoral Côte D'Opale, IRD, UMR 8187, LOG—Laboratoire D'Océanologie et de Géosciences, Station Marine de Wimereux, F- 59000 Lille, France. [8]Leibniz Institute of Freshwater Ecology and Inland Fisheries, IGB, 12587 Berlin, Germany. [9]Freie Universität Berlin, Department of Biology, Chemistry and Pharmacy, 14195 Berlin, Germany. [10]National Research Institute for Agriculture, Food and Environment (INRAE), CARRTEL, Université Savoie Mont Blanc, 74200 Thonon les Bains, France. [11]Centre for Limnology, Institute of Agricultural and Environmental Sciences, Estonian University of Life Sciences, Kreutzwaldi 5D, 51014 Tartu, Estonia. [12]Department of Ecosystem Studies, School of Environmental Science, The University of Shiga Prefecture, Hikone 522-8533 Shiga, Japan. [13]Kinneret Limnological Laboratory, Israel Oceanographic & Limnological Research, P.O. Box 447, 14950 Migdal, Israel. [14]Biodiversity Research Center, Academia Sinica, Taipei 11529, Taiwan. [15]UK Centre for Ecology & Hydrology, Lancaster Environment Centre, Library Avenue, Bailrigg, Lancaster, Lancashire LA1 4AP, UK. [16]Lake Biwa Environmental Research Institute, Otsu 520-0022, Japan. [17]Faculty of Environment and Information Sciences, Yokohama National University, Yokohama 240-8502 Kanagawa, Japan. [18]Department of Environmental Science, Faculty of Science, Toho University, Funabashi, Chiba 274-8510, Japan. [19]Research Center for Lake Biwa & Environmental Innovation, Ritsumeikan University, Kusatsu 525-0058 Shiga, Japan. [20]Biodiversity Division, National Institute for Environmental Studies, 16-2 Onogawa, Tsukuba, Ibaraki 305-8506, Japan. [21]CNR Water Research Institute (IRSA), L.go Tonolli 50, 28922 Verbania, Pallanza, Italy. [22]Plymouth Marine Laboratory, Prospect Place, West Hoe, Plymouth PL1 3DH, UK. [23]Institute of Ecology and Evolutionary Biology, Department of Life Science, National Taiwan University, Taipei 10617, Taiwan. ✉email: chsieh@ntu.edu.tw

