## [Peer Review File · Nature Communications]

Reviewers' Comments:

Reviewer #1:

Remarks to the Author:

The study of phytoplankton production has largely been studied from the point of view of eutrophication of waterbodies, so that largely unidirectional effects have been studied, phosphorus (P) → EF (production) and nitrogen (N) → EF. Diversity has been less of a focus in such studies, as models seem to show that a few phytoplankton functional types (stoichiometries in P and N concentrations) give a good approximation of phytoplankton production. In contrast to such studies of phytoplankton production prompted by concerns about eutrophication and in which there are strong external nutrient loading drivers, the authors here focus on oligotrophic waterbodies and biodiversity (BD). Here species richness, is a main focus along with production and nutrients N and P.

As the authors note, the BDEF relationship has been studied more frequently in terrestrial than in aquatic systems, as studied here (Page 8), although some studies are noted by Waide et al. (1999, ARES). More novel is that this is the first application of convergent cross mapping (CCM) that I know of that has been applied to study causal loops among these variables in aquatic ecosystems. Therefore, the study seems to break new ground in BDEF relationships. A recent terrestrial has shown evidence that, at least in grasslands, effects of BD on EF are seen in real-world systems, not just experiments based on random assembly (Jochum et al. 2020 Nature ecology & evolution). The current study also uses real-world aquatic systems, which have been monitored over long time periods (16-19 years), so the results here seem to be a reasonable parallel to the recent grassland ecosystem results.

One question is whether this study has some implications for the more applied research on eutrophication in waterbodies. The authors state the "effect of plant diversity on production can be substantially underestimated if feedbacks between plant production and the environment are overlooked" (Page 6). Does this show that neglect of feedbacks between plant production and the environment substantially underestimate production, which is suggested if species richness is a main causal agent in phytoplankton production? This may depend on the degree to which the results of the present study are limited to oligotrophic systems.

One of the results found by the authors is that NO₃, along with species richness, was the main controlling factor on phytoplankton production (Page 9). How does that result relate to the long-standing view of phosphorus as the main limiting nutrient of production in temperate and boreal lakes (e.g., Schindler 1978, Limnology and Oceanography) and more recent research indicating colimitation of N and P (Paerl et al. 2016, Environmental Science and Technology, Lewis et al. 2020, Inland Waters)?

The authors found a strong effect of species richness on phytoplankton production, similar to the cited study of Duffy et al. (2017), Does the analysis here shed any light on the question of whether the effects of biodiversity on ecosystem function are results of species complementarity (Tilman et al. 1997) or selection effect (i.e., higher probability of occurrence of dominant species or traits that would have a strong effect on ecosystem function; Huston 1997, Oecologia, Grime 1998 J. Ecology)?

The authors note that "primary production was a significant driver of phytoplankton species richness in most ecosystems" (Page 10). Some studies have shown positive relationships for the effects of productivity on species richness and some have shown unimodal relationships. Is it likely that in the studies analyzed here the productivity levels here low enough that the decreasing branch of species richness vs. production is not reached?

The authors found that systems with lower average temperature had stronger BD → EF effect (Page 12). Is this possibly related to the indication that higher temperature leads to stronger interspecific competition and causes the relationship to become flat or even negative for high temperature (Parain et al. 2019, The American Naturalist)? If true, although "greater diversity help

mitigate the effects of warming" (Page 13), warming may decrease the BDEF effects.

The authors note that "Our findings contrast with the prevailing view that ecosystem functioning was insensitive to changes in diversity at high levels of diversity." (Page 14). That is an interesting finding, as simulation models examining eutrophication often have only a small number of phytoplankton variables, though those are often functional types (e.g., 3 phytoplankton groups in PCLake, Mooij et al. 2007, Hydrobiologia). Are the functional types in the present study well enough known to determine what a minimum number of functional types is needed to approximate total phytoplankton production? Concerning species richness, 10 phytoplankton species are represented in the BLOOM II model (Los, 1980, Hydrobiological Bulletin). How does that compare with the species richness in the aquatic ecosystems studied here?

The authors compare the CCM approach with structural equation modeling, SEM (Page 20). It is clear that CCM has an advantage of being applicable to general nonlinear systems in which not all relevant variables are known. However, SEM has been very useful in teasing apart the paths by which species richness can affect community production, through various types of functional diversity, species traits, and environmental conditions, thus allowing mechanisms to be determined. Ideally, CCM and SEM could be used together, as they are complementary approaches.

The strength of the current study is that is a new application of CCM that allows detection of feedback causality in natural phytoplankton communities. The weakness is that, by itself, it does not reveal the mechanisms that are underlying these feedbacks. But despite that weakness, the study reveals some interesting new results, such as strong nitrate-mediated feedbacks in warm ecosystems and phosphate-mediated feedbacks in cold ecosystems. This will be of interest to ecologists interested in the diversity-ecosystem function issue.

Minor

Page 8, Line 8. Change 'are mainly' to 'is mainly'

Page 10, two bottom lines. Change 'is more often operated at short-term scales' to 'more often operates at short-term scales'

Page 13, Line 5 from bottom. 'productivity (phytoplankton biomass as a proxy)'. I thought that chl-a was the proxy used.

Reviewer #2:

Remarks to the Author:

Review on Causal networks of phytoplankton diversity and production are modulated by environmental context by Chang et al.

This paper analyses how biodiversity and environmental factors modulate ecosystem services. The work differentiates from previous approaches by using a network approach instead of different types of regressions that already imply a causality and deny the potential feedback effects. They have used this approach on an extensive database of phytoplankton time series from different regions of the world. Overall I believe this is methodologically a very strong paper, proposing an innovative approach to study feedback effects on ecosystems. However I have major issues with the use and interpretation of variables that in my view make difficult to evaluate if the results are valid or not.

Major issues:

Chl a as a proxy of production: Even if Chl a was a good proxy of biomass (B, see below), in this context it cannot be taken as a proxy of production (P). Precisely, for the same existing biomass different environmental conditions (nutrients, light, temperature) will increase or decrease production (by changing growth rates or productivity). Therefore the approach taken here is downplaying the role of the environment by definition, because using a proxy of biomass as a proxy of production. In a very large scale comparison, when comparing different ecosystems, biomass could be taken as a rough proxy of production, as B will have a significant effect on P. But

when analysing in each of the ecosystems subtle effects such as diversity and environment on P, by considering B=P you are specifically downplaying one of the factors (environment) you want to analyse.

Chl a as proxy of biomass: In this context I even have problems considering Chl a as a useful proxy of biomass. Chl a:C ratio can vary widely depending on environmental conditions. A summer dinoflagellates bloom under high irradiance is likely to have a lower chl a signal than an early spring diatom one, but similar or higher biomass. Again, in a rough large scale comparison we can assume higher Chl a higher biomass, but when looking at a feedback effect in a time series the Chl a -> biomass -> production series of assumptions is stretching the data to the extreme.

Diversity: It is difficult to evaluate without looking at the raw data of each time series, however I would argue that richness is not the best diversity index for this analysis. In terms of richness a phytoplankton bloom is not a major change in number of species (some large species more abundant), whereas in terms of evenness or Shannon you can see a clear signal of one or a few species dominating the ecosystem.

Data preparation: For the time series with a higher than monthly resolution data are averaged (also more than one depth). I do not understand what is the purpose of this operation and I think it introduces a significant bias. Time series are not compared to each other, so I do not see the objective of the average. By doing that in some of the time series the analysis corresponds to a very specific observation whereas in others is an average of conditions. Considering the time scale of phytoplankton growth, in one you will have peaks because sampling coincided with a specific event, whereas in the other time series you will never get such a situation because you are averaging all the month conditions and smoothing all the peaks.

Environmental factors: The objective of the paper is to modulate the effect of environmental factors and diversity on production. However, the environmental factors considered are limited to nutrients and temperature. If it is production other basic environmental factors such as irradiance, euphotic depth and water column stability should be included in the analysis. Otherwise we are excluding some of the main drivers of production from the analysis, which affects conclusions.

Overall, I would say this is a very strong paper on the methodological side, but the conclusions cannot be granted due to the extreme assumptions taken with the variables.

Reviewer #3:

Remarks to the Author:

The article main claim is well presented in the abstract and title. It is that, concerning phytoplankton in aquatic bodies, the link between diversity and functioning, here primary production, is modulated by the environmental context. I found the demonstration of this claim outstanding. This demonstration is based on an exceptional data set of long time series in 19 aquatic systems. The analysis is then based on a numerical approach combining a perfectly complementary set of data analyses, combining pretty recently developed ones with more classical ones. Such results are of great interest in the community of ecology, with a wider possible impact more globally in biology, as it concerns a nested set of levels of organization, from the phytoplankton cell to the whole ecosystem, as the primary production is highly determinant for the whole functioning of the food web and other networks of interactions connected to the food web (forming what we call now a multiplex). An important output of this work will be to give the readers a new look on former Biodiversity / Functioning studies. By demonstrating that looking for generalities may not be our main objective in ecological studies, the authors should reorient the future protocols of data analysis in this domain. Another consequence of the authors' conclusion is to find a nice explanation of the apparent contradiction found previously in papers attending to join biodiversity and functioning, especially in the domain of plankton studies.

Concerning the statistics chosen, I consider that the choices made in the set of statistical analyses and associated parameters is perfectly appropriate. I just have a few comments, presented hereafter, that may help improving the readability of their presentation.

1) Better explain within the text the difference in considering endogenous vs exogenous

constraints and implications of such choices in terms of feedback loops. In fact, these explanation are very good in the figure 1 legend, but I think it is missing from the text of the introduction.

2) Better explain the way the phytoplankton species richness is determined, if there was any process of quality control, in order to ensure the comparability of the data sets. I also missed the information about how the level of the species determination was reached, e.g. was electronic microscopy added, when needed, to optic microscopy.

3) Please explain further how chl a concentrations were translated into production values.

4) I also missed information on the choice of CCA rather than RDA in the multivariate analyses.

The connection between explanatory and explained variables would not follow the same model in the two cases, and the choice of CCA is probably justified, but the confirmation should be given in the text. What is the justification of the choice of a modal rather than linear model in the relationship? Did you use coenoclines to drive this choice?

5) Please give the number of iterations of the permutation tests.

Comments from reviewers are copied below with our point-by-point responses. We use black text for the reviewer's comments and blue text for our responses. Changes in the text of the revised manuscript are underlined.

Reviewer #1:

The study of phytoplankton production has largely been studied from the point of view of eutrophication of waterbodies, so that largely unidirectional effects have been studied, phosphorus (P) → EF (production) and nitrogen (N) → EF. Diversity has been less of a focus in such studies, as models seem to show that a few phytoplankton functional types (stoichiometries in P and N concentrations) give a good approximation of phytoplankton production. In contrast to such studies of phytoplankton production prompted by concerns about eutrophication and in which there are strong external nutrient loading drivers, the authors here focus on oligotrophic waterbodies and biodiversity (BD). Here species richness, is a main focus along with production and nutrients N and P.

Response:

Thank you for this excellent summary of our work. We appreciate all the constructive comments from Reviewer #1, and the insightful discussion about our manuscript. However, we wish to clarify that the aquatic systems analyzed in this study comprised not only oligotrophic but also mesotrophic as well as eutrophic systems. Averaged concentration of total phosphorus (TP) ranged from 7.5 to 126.4 $\mu\text{g/L}$ in the freshwater systems analyzed. No TP measures were available in the two marine stations [Western English Channel (Wc) and Narranganset Bay (Ng)]. In the revision, we included the information on TP in the updated Table S1 and clarified the main text at **P8:L13-17**, "Therefore, we employed CCM³³ to assemble causal networks for 19 sites (Fig. S1) among 16 globally distributed ecosystems (Methods; Fig. S1), representing various types of aquatic ecosystems with various morphometrics and trophic states (from oligotrophic to eutrophic systems presented in Table S1)."

Comment 1:

As the authors note, the BDEF relationship has been studied more frequently in terrestrial than in aquatic systems, as studied here (Page 8), although some studies are noted by Waide et al. (1999, ARES). More novel is that this is the first application of convergent cross mapping (CCM) that I know of that has been applied to study causal loops among these variables in aquatic ecosystems. Therefore, the study seems to break new ground in BDEF relationships. A recent terrestrial has shown evidence that, at least in grasslands, effects of BD on EF are seen in real-world systems, not just experiments based on random assembly (Jochum et al. 2020 Nature ecology &

evolution). The current study also uses real-world aquatic systems, which have been monitored over long time periods (16-19 years), so the results here seem to be a reasonable parallel to the recent grassland ecosystem results.

Response 1:

We agree with the reviewer's opinion that our study firstly revealed various causal feedback loops in aquatic systems based on CCM analysis of long-term time series data. Indeed, our analysis presented a reasonable parallel to the findings in grassland systems suggested by the reviewer (Jochum et al. 2020 Nature Ecology & Evolution). Therefore, we included this reference (Ref_61) and highlighted this issue at

P18:L19-21, "The selected long-term datasets from various aquatic ecosystems represented a reasonable parallel to long-term biodiversity experiments conducted in terrestrial grassland ecosystems⁶¹."

Comment 2:

One question is whether this study has some implications for the more applied research on eutrophication in waterbodies. The authors state the "effect of plant diversity on production can be substantially underestimated if feedbacks between plant production and the environment are overlooked" (Page 6). Does this show that neglect of feedbacks between plant production and the environment substantially underestimate production, which is suggested if species richness is a main causal agent in phytoplankton production? This may depend on the degree to which the results of the present study are limited to oligotrophic systems.

Response 2:

As explained above, our analysis included also eutrophic waterbodies. Thus, the conclusion obtained from this study can be also applied in eutrophic systems. Considering the sentence pointed out by the reviewer, we suggested that influences of diversity may be underestimated if diversity-mediated feedbacks were overlooked. However, neglect of feedbacks might not necessarily lead to underestimation of plant production unless influences of feedbacks are always positive. Nonetheless, the sign of feedbacks was not determined by CCM, as we caveated at **P17:L8-12**. To avoid any confusion, we revised the sentence at **P6:L7-9**, "For example, impacts of plant diversity loss on plant biomass cannot be precisely evaluated if feedbacks among plant diversity, biomass and environment are overlooked⁸."

Comment 3:

One of the results found by the authors is that NO₃, along with species richness, was the main controlling factor on phytoplankton production (Page 9). How does that result relate to the long-standing view of phosphorus as the main limiting nutrient of

production in temperate and boreal lakes (e.g., Schindler 1978, Limnology and Oceanography) and more recent research indicating colimitation of N and P (Paerl et al. 2016, Environmental Science and Technology, Lewis et al. 2020, Inland Waters)?

Response 3:

The importance of nitrogen supply is a seemingly counter-intuitive result with respect to the long-standing view that phosphorus is the main limiting nutrient in many freshwater systems. However, we suggest the importance of nitrogen in this study may not be surprising, as phosphorus was not always a limiting factor in the systems we analyzed, e.g., Lake Geneva (Gv) (Anneville et al., 2002 Limnology and Oceanography) and Lake Kasumigaura (Ks) (Matsuzaki et al. 2018, Ecology). Moreover, nitrogen deficiency controlled the magnitude of summer cyanobacteria blooms in Müggelsee (Shatwell and Köhler 2019, Limnology and Oceanography). In these cases, fluctuations in phytoplankton biomass would not necessarily respond to changes in phosphate concentration, as many of our analyzed systems were P-rich. For example, Lakes Mendota (Me), Lake Monona (Mo) and Lake Müggelsee (Mu), have high average phosphate concentrations = 57.5, 41.7, and 65.9 $\mu\text{g/L}$, respectively (Table S1). In addition, there was abundant total phosphorus in shallow lake systems (average TP were 106.1, 112.5, and 126.5 $\mu\text{g/L}$ in Lake Inba (Ib), Lake Kasumigaura (Ks), and Müggelsee (Mu), respectively).

However, we agree with the reviewer's proposal that colimitation by N and P might be important and help to explain our findings, although this warrants more detailed investigations. To clarify this issue, we included our explanation and the citation of colimitation in the main text at **P10:L22-P11:L9**, “The importance of NO_3 uncovered in our analyses might not be a counter-intuitive result, as many systems analyzed in this study were P-rich. For instance, the average phosphate concentration was 57.5 and 41.7 $\mu\text{gP/L}$ for Lakes Mendota (Me) and Lake Monona (Mo) (Table S1), respectively. In addition, there was also high total phosphorus (TP) in shallow lake systems, e.g., average TP was 106.1, 112.5, and 126.4 $\mu\text{gP/L}$ in Lake Inba (Ib), Lake Kasumigaura (Ks), and Müggelsee (Mu), respectively. Phosphorus was not always a limiting factor in eutrophic and mesotrophic systems, e.g., Lake Kasumigaura³⁹ and Lake Geneva (Gv)⁴⁰. In addition, nitrogen was deficient and limited cyanobacteria bloom in Müggelsee (Mu)⁴¹. Nonetheless, we cannot exclude the possibility of colimitation⁴² in N and P and the possibility that P availability also depends on N^{43} , which warrants further investigation.”

Ref_39: Matsuzaki, S.-i. S., Suzuki, K., Kadoya, T., Nakagawa, M. & Takamura, N. Bottom-up linkages between primary production, zooplankton, and fish in a shallow, hypereutrophic lake. *Ecology* 99, 2025-2036 (2018).

Ref_40: Anneville, O. et al. Temporal mapping of phytoplankton assemblages in Lake Geneva: annual and interannual changes in their patterns of succession. *Limnology and Oceanography* 47, 1355-1366 (2002).

Ref_41: Shatwell, T. & Köhler, J. Decreased nitrogen loading controls summer cyanobacterial blooms without promoting nitrogen-fixing taxa: Long-term response of a shallow lake. *Limnology and Oceanography* 64, S166-S178 (2019).

Ref_42: Paerl, H. W. et al. It takes two to Tango: when and where dual nutrient (N & P) reductions are needed to protect lakes and downstream ecosystems. *Environmental Science & Technology* 50, 10805-10813 (2016).

Ref_43: Schindler, David W. The dilemma of controlling cultural eutrophication of lakes. *Proceedings of the Royal Society B: Biological Sciences* 279, 4322-4333 (2012).

Comment 4:

The authors found a strong effect of species richness on phytoplankton production, similar to the cited study of Duffy et al. (2017), Does the analysis here shed any light on the question of whether the effects of biodiversity on ecosystem function are results of species complementarity (Tilman et al. 1997) or selection effect (i.e., higher probability of occurrence of dominant species or traits that would have a strong effect on ecosystem function; Huston 1997, Oecologia, Grime 1998 J. Ecology)?

Response 4:

Unfortunately, the causal strength of phytoplankton diversity derived from CCM cannot be further decomposed to species complementarity or selection effects. Theoretically, it is possible to mechanistically investigate various biodiversity effects on ecosystem functioning if mechanistic measures (some sort of indices) indicating species complementarity or selection effect can be derived and monitored in natural systems. If so, we could apply CCM to quantify the strength of their causal effects on ecosystem functioning and then determine their relative importance. However, to our best knowledge, the measurements that would allow us to differentiate niche complementarity or selection effects were mostly computed using experimental data (e.g., the method presented in Ref_60: Loreau and Hector 2001, *Nature*) but remain a challenging task for observational data. Therefore, distinguishing complementarity versus selection effect is beyond the scope of our study. However, we still caveat this issue in the Discussion at **P18:L11-16**, “For instance, biodiversity was suggested to influence ecosystem functioning via species complementarity or selection effects.”

Measuring complementarity or selection effects is available for experimental data⁶⁰, but remains a challenging task for observational data. Thus, incorporation of these detailed mechanistic measures in the causal networks is an important future research topic.”

Ref_60: Loreau, M. & Hector, A. Partitioning selection and complementarity in biodiversity experiments. Nature 412, 72-76 (2001).

Comment 5:

The authors note that "primary production was a significant driver of phytoplankton species richness in most ecosystems" (Page 10). Some studies have shown positive relationships for the effects of productivity on species richness and some have shown unimodal relationships. Is it likely that in the studies analyzed here the productivity levels here low enough that the decreasing branch of species richness vs. production is not reached?

Response 5:

CCM is a nonparametric approach that makes no assumptions about the functional form (e.g., linear or unimodal) of the relationship between phytoplankton diversity and biomass. This equation-free property is important for statistical inferences on nonlinear dynamical systems wherein governing equations are highly complicated and quantitative relationships between variables change with context. Indeed, phytoplankton species richness had a variety of relationships with primary productivity, depending on environmental contexts (e.g., patterns revealed in Ref_X1: Dodson et al. 2000, Ecology). Applying CCM enabled us to reveal the causal effects of phytoplankton biomass on diversity even if the true relationship was unimodal or of a more complicated form. We clarified this issue in a new paragraph in Methods at **P22:L22-P23:L9**, “In addition, CCM is a nonparametric approach, free from assumptions of specific form of quantitative relationships between causal variables. Although this makes CCM difficult to explore quantitative features, e.g., the minimal number of species required to maintain 80% levels of ecosystem function, it provides high flexibility to infer causations in nonlinear dynamical systems. Such flexibility is important for inferring nonlinear dynamical systems, because quantitative relationships between any two dynamical variables could change, depending on the varying state of other state variables⁷⁵ or environmental contexts. For example, linear associations between two variables will appear then disappear or change sign – so-called mirage correlations³³, making methods based on modelling static, parametric relationships difficult to correctly identify causations⁹.”

Ref_X1: Dodson, S. I., Arnott, S. E., & Cottingham, K. L. (2000). The relationship in lake communities between primary productivity and species richness. *Ecology* 81, 2662-2679 (2000).

Comment 6:

The authors found that systems with lower average temperature had stronger BD → EF effect (Page 12). Is this possibly related to the indication that higher temperature leads to stronger interspecific competition and causes the relationship to become flat or even negative for high temperature (Parain et al. 2019, *The American Naturalist*)? If true, although "greater diversity help mitigate the effects of warming" (Page 13), warming may decrease the BDEF effects.

Response 6:

Yes, the comment made by the reviewer is consistent with our findings. We agree with the reviewer's interpretation, suggesting that warming weakens diversity effects because of strong interspecific competition under high temperature. Therefore, we included this citation (Ref_48: Parain et al. 2019, *The American Naturalist*) to the main text and highlight this issue at **P14:L7-10**, "Perhaps greater diversity and its effects mitigate adverse impacts of global warming⁹ although warming may also weaken biodiversity effects on ecosystem functioning due to strong interspecific competitions under high temperatures⁴⁸."

Comment 7:

The authors note that "Our findings contrast with the prevailing view that ecosystem functioning was insensitive to changes in diversity at high levels of diversity." (Page 14). That is an interesting finding, as simulation models examining eutrophication often have only a small number of phytoplankton variables, though those are often functional types (e.g., 3 phytoplankton groups in PCLake, Mooij et al. 2007, *Hydrobiologia*). Are the functional types in the present study well enough known to determine what a minimum number of functional types is needed to approximate total phytoplankton production? Concerning species richness, 10 phytoplankton species are represented in the BLOOM II model (Los, 1980, *Hydrobiological Bulletin*). How does that compare with the species richness in the aquatic ecosystems studied here?

Response 7:

The merit of applying BLOOM is that phytoplankton taxa have two variants: light sensitive and nutrient sensitive. Thus, BLOOM allows the resolution of performance variability within functional categories of taxa. However, 10 "species" is good only when the main species in ecosystems are represented. As an example, BLOOM doesn't have any species with *Mougeotia gracillima* characteristics. Consequently,

BLOOM produces bad results when *Mougeotia* dominate. This is what we found in Lake Geneva (**Ref_X2**: Soullignac *et al.* 2019, Knowl. Manag. Aquat. Ecosyst.).

In our meta-analysis, the average species richness in those systems ranged from 15.1-38.9 with high turnover rates (Table S1). In addition, the total number of species was much higher compared to the species richness used in ecosystem models like BLOOM. Despite including numerous species, CCM, a nonparametric approach as previously explained, does not aim to reconstruct a static parametric relationship. Without reconstructing a parametric relationship between ecosystem functioning and species richness (i.e., BDEF relationship), the minimal species richness required to achieve a certain ecosystem function cannot be obtained from the current analysis. Nonetheless, we suggest that a strong causal effect of species richness on phytoplankton biomass revealed by CCM indicates that the dynamics of phytoplankton biomass responded strongly to changing species richness. Moreover, the strength of causal effects substantially varied with environmental contexts (Fig. 4), implying that quantitative BDEF relationships as well as the minimal required species arising from the causations between diversity and ecosystem functioning also highly depend on varying environmental contexts in natural systems. We clarify and caveat this issue at **P22:L22-P23:L4**, “In addition, CCM is a nonparametric approach, free from assumptions of specific form of quantitative relationships between causal variables. Although this makes CCM difficult to explore quantitative features, e.g., the minimal number of species required to maintain 80% levels of ecosystem function, it provides high flexibility to infer causations in nonlinear dynamical systems.” We also include the interpretation of causal strength in Methods at **P24:L8-13**, “As in previous studies^{74,77}, the strength of causal interaction was quantified based on the cross-mapping skill at the maximal library length, $\rho(L_{\max})$. That is, stronger causal effects result in convergence to high cross-map skill^{9,33,77}. For instance, a strong causal effect of species richness on phytoplankton biomass revealed by CCM indicated that dynamics of phytoplankton biomass (magnitude or variability) responded strongly to changing species richness.”

Ref_X1: Soullignac F., Anneville O., Bouffard D., Chanudet V., Dambrine E., Guénand Y., Harmel T., Ibelings B., Trevisan D., Uittenbogaard R., Danis P.-A. Contribution of 3D coupled hydrodynamic-ecological modeling to assess the representativeness of a sampling protocol for lake water quality assessment. Knowledge & Management of Aquatic Ecosystems 420, 42 (2019).

Comment 8:

The authors compare the CCM approach with structural equation modeling, SEM (Page 20). It is clear that CCM has an advantage of being applicable to general nonlinear systems in which not all relevant variables are known. However, SEM has been very useful in teasing apart the paths by which species richness can affect community production, through various types of functional diversity, species traits, and environmental conditions, thus allowing mechanisms to be determined. Ideally, CCM and SEM could be used together, as they are complementary approaches.

Response 8:

We are sorry for causing the misunderstanding, due to our inappropriate comparison between CCM and SEM in the previous version. In fact, what we really argued is that CCM enables implicit incorporation of the influences of other confounding variables using embedded time-lags. Although SEM might be useful in teasing apart paths by which one variable can affect others, SEM might not be a suitable method to tackle the datasets we analyzed, as SEM assumes static quantitative relationships between variables. Although this assumption can be fulfilled in linear stochastic systems, this strong assumption is not valid in nonlinear dynamical systems. For example, quantitative relationships between species richness and phytoplankton biomass were time-varying in most systems, switching between various functional forms (linear or unimodal) as well as the sign of relationship (positive or negative; i.e., mirage correlation). Fig. X1 below was adopted from Fig. S14 in a recent study (Chang et al. 2020, Global Change Biology) and the other systems additionally analyzed in this study are presented in Fig. X2 based on the open-source code https://github.com/biozoo/GCB_SI_Rscript. These figures indicate highly variable forms of quantitative relationships between phytoplankton species richness and biomass, which apparently deviated from the assumption of static relationship in SEM. Due to violations of the critical assumption, we did not use SEM to investigate the causal mechanisms despite SEM being a powerful method in linear stochastic systems. To avoid any confusion, we only included in Fig. X1-2 in this response letter. We discarded the comparison between CCM and SEM from Methods. We made clarification at **P22:L16-22**, “CCM analysis accounts for influences of confounding variables implicitly. Specifically, CCM incorporates influences of confounding variables using lagged embeddings, e.g., $(X_{t-1}, X_{t-2}, \dots)$, which have accounted for historical effects of other variables using lagged terms, even if those variables were unobserved or difficult to identify. As such, CCM does not require identifying or ruling out influences of confounding variables in order to quantify causations between two variables, and thus can be applied in more general dynamical systems⁶⁴.”

Fig. X1. Mirage correlations between species richness and phytoplankton biomass in 10 ecosystems. The model was selected based on AIC. The black line indicated dynamics of linear regression coefficient l . However, if the quadratic model was selected ($AIC_{\text{quadratic}} < AIC_{\text{linear}}$), quadratic coefficient q was shown instead using a blue line and a solid black line (linear regression coefficient) was changed to a dashed line. (adopted from Fig. S14 in Chang et al. 2020, *Global Change Biology*)

Fig. X2 Mirage correlations between species richness and phytoplankton biomass in the additional nine ecosystems not included in Fig. X1.

Comment 9:

The strength of the current study is that is a new application of CCM that allows detection of feedback causality in natural phytoplankton communities. The weakness is that, by itself, it does not reveal the mechanisms that are underly these feedbacks. But despite that weakness, the study reveals some interesting new results, such as strong nitrate-mediated feedbacks in warm ecosystems and phosphate-mediated feedbacks in cold ecosystems. This will be of interest to ecologists interested in the diversity-ecosystem function issue.

Response 9:

Thank you for pointing out the novelty of our study. However, we hope to use CCM to address the issue concerning the resolution of causal mechanisms. In theory, it is still possible to evaluate the importance of various causal paths (e.g., averaged strength of a causal path) if time series data of detailed mechanistic measures (e.g., an index indicating selection effects) are available, as explained in Response 4. Although very detailed measurements might be difficult to obtain across numerous systems, we are looking forward to seeing the application of our proposed methodological framework to explore more detailed causal mechanisms in some systems where comprehensive measurements are available. Thus, we caveat this issue in the Discussion at **P18:L8-17**, “It is noteworthy that our proposed methodological

framework can be applied to explore more detail causal feedbacks or paths if precise mechanistic measures (e.g., nutrient recycling rate rather than nutrient stock) can be monitored through time. For instance, biodiversity was suggested to influence ecosystem functioning via species complementarity or selection effects. Measuring complementarity or selection effects is available for experimental data⁶⁰, but remains a challenging task for observational data. Thus, incorporation of these detailed mechanistic measures in the causal networks is an important future research topic. More comprehensive surveys are required in future ecological monitoring to improve our understanding of causal mechanisms embedded in causal networks.”

Minor

Comment 10:

Page 8, Line 8. Change 'are mainly' to 'is mainly'

Response 10:

Corrected at **P8:L9**.

Comment 11:

Page 10, two bottom lines. Change 'is more often operated at short-term scales' to 'more often operates at short-term scales'

Response 11:

Corrected at **P11:L20-21**.

Comment 12:

Page 13, Line 5 from bottom. 'productivity (phytoplankton biomass as a proxy)'. I thought that chl-a was the proxy used.

Response 12:

Following the comments from reviewer #2, we used chlorophyll *a* concentration (Chl*a*) as a proxy of phytoplankton biomass instead of a proxy of productivity. Therefore, we revised this sentence at **P14:L19-20** as, “Water temperature and phytoplankton biomass (Chl*a* as a proxy) level are also critical to determining strengths of various pairwise feedbacks.”

Thank you very much for these comments that greatly improved our manuscript.

Reviewer #2

Review on Causal networks of phytoplankton diversity and production are modulated by environmental context by Chang et al.

This paper analyses how biodiversity and environmental factors modulate ecosystem services. The work differentiates from previous approaches by using a network approach instead of different types of regressions that already imply a causality and deny the potential feedback effects. They have used this approach on a extensive database of phytoplankton time series from different regions of the world. Overall I believe this is methodologically a very strong paper, proposing an innovative approach to study feedback effects on ecosystems. However I have major issues with the use and interpretation of variables that in my view make difficult to evaluate if the results are valid or not.

Response:

Thank you for pointing out the novelty as well as methodological strength in this work. Following the reviewer's comments, we carefully adjusted the interpretation of variables and provided detailed justifications for the variables we analyzed.

Major issues:

Comment 1:

Chl *a* as a proxy of production: Even if Chl *a* was a good proxy of biomass (B, see below), in this context it cannot be taken as a proxy of production (P). Precisely, for the same existing biomass different environmental conditions (nutrients, light, temperature) will increase or decrease production (by changing growth rates or productivity). Therefore the approach taken here is downplaying the role of the environment by definition, because using a proxy of biomass as a proxy of production. In a very large scale comparison, when comparing different ecosystems, biomass could be taken as a rough proxy of production, as B will have a significant effect on P. But when analysing in each of the ecosystems subtle effects such as diversity and environment on P, by considering B=P you are specifically downplaying one of the factors (environment) you want to analyse.

Response 1:

We acknowledge the problems in adopting chlorophyll *a* concentration (Chl*a*) as a proxy of production. Firstly, we hope to clarify that our aim in this manuscript is to present a novel methodological framework to investigate causal feedbacks in natural ecosystems. Thus, we agree with the reviewer's comment about avoiding an overly strong assumption regarding the use of Chl*a* as a proxy for production. However, we

still consider Chl_a as a valid proxy for phytoplankton biomass and replaced the wording of ‘phytoplankton production’ with ‘phytoplankton biomass’ throughout this manuscript. To justify using Chl_a as a proper proxy for phytoplankton biomass, we provide a detailed explanation in the following response.

Comment 2:

Chl a as proxy of biomass: In this context I even have problems considering Chl a as a useful proxy of biomass. Chl a:C ratio can vary widely depending on environmental conditions. A summer dinoflagellates bloom under high irradiance is likely to have a lower chl_a signal than an early spring diatom one, but similar or higher biomass. Again, in a rough large scale comparison we can assume higher Chl a higher biomass, but when looking at a feedback effect in a time series the Chl a -> biomass -> production series of assumptions is stretching the data to the extreme.

Response 2:

We understand that the Chl_a:C ratio is not a constant in many situations. However, in the meta-analysis to compare the lakes at a large-scale, Chl_a was the most available and important measure indicating phytoplankton standing stock in various types of aquatic systems. Here, we suggested that Chl_a concentration effectively represented phytoplankton community biomass in our case. Indeed, there were strong positive relationships between Chl_a and phytoplankton community biomass (Fig. X3) calculated from composition data (total sum of individual species biomass). Such a positive relationship held in all systems, except for Feitsui Reservoir (Ft) in which small phytoplankton cells were not effectively collected by the 10-um plankton net. We avoided using total biomass data, derived from our composition dataset, because this measure is statistically dependent on our diversity measures (also derived from composition data). In addition, the use of Chl_a as a proxy of algal biomass is widely applied in the BDEF literature (e.g., Cardinale 2011, Nature & Lewandowska *et al.* 2016, Philos. Trans. R. Soc. B). Thus, we only presented this Fig. X3 in the response letter and included these citations in Methods at **P20:L7-9**, “(2) chlorophyll-a concentration as a measure of phytoplankton biomass and ecosystem function, a widely used proxy of algal biomass in the BDEF literature^{17,62}”

Ref_17: Cardinale, B. J. Biodiversity improves water quality through niche partitioning. Nature 472, 86-89, (2011).

Ref_62: Lewandowska, A. M. et al. The influence of balanced and imbalanced resource supply on biodiversity–functioning relationship across ecosystems. Philosophical Transactions of the Royal Society B: Biological Sciences 371, (2016).

Fig. X3 Statistical relationships between chlorophyll *a* concentration and community biomass derived from composition data (both were log transformed and normalized) for each system. In most systems, chlorophyll *a* concentration was highly correlated with community biomass derived from composition data (mean correlation coefficient= 0.668, $p < 0.001$ for all systems except for Ft ($p = 0.76$)).

Comment 3:

Diversity: It is difficult to evaluate without looking at the raw data of each time series, however I would argue that richness is not the best diversity index for this analysis. In terms of richness a phytoplankton bloom is not a major change in number of species (some large species more abundant), whereas in terms of evenness or Shannon you can see a clear signal of one or a few species dominating the ecosystem.

Response 3:

We analyzed Shannon diversity and determined that results based on Shannon diversity were consistent with the findings based on species richness, indicating the importance of nutrients. However, the causal effects from phytoplankton biomass became relatively more important compared to species richness (Fig. S5a). This result is reasonable because changes in total phytoplankton biomass might be better associated with dominance of some phytoplankton species, which in turn influences those diversity indices considering species relative composition. Therefore, we included the analyses of Shannon diversity in the Supplemental materials, Fig. S6a and addressed our findings in main text at **P10:L13-17**, “The results suggest that the importance of nutrients is robust to the use of other diversity index (e.g., Shannon diversity in Fig. S5) although the causal effects from phytoplankton biomass became relatively more important compared to biomass effects on species richness (Fig. 2b).”

However, the effects of Shannon diversity on phytoplankton biomass were on average weaker than the effects of species richness (Fig. S5b). Indeed, there is no consensus about whether Shannon diversity or other evenness indices are more important to ecosystem functioning than species richness. At least in some recent studies listed below (Ref_X3-5), effects of Shannon diversity (or another evenness index) on ecosystem functioning (e.g., biomass) were not more important than effects of species richness. Therefore, we only include the findings of Shannon diversity in Supplemental materials.

Ref_X3: Yi, S. et al. (2021) Biodiversity, environmental context and structural attributes as drivers of aboveground biomass in shrublands at the middle and lower reaches of the Yellow River basin. *Science of The Total Environment* 774, 145198.

Ref_X4: Li, Y. et al. (2019) Relationships between plant diversity and biomass production of alpine grasslands are dependent on the spatial scale and the dimension of biodiversity. *Ecological Engineering* 127, 375-382.

Ref_X5: Sanaei, A., Ali, A., Chahouki, M. A. Z. & Jafari, M. (2018) Plant coverage is a potential ecological indicator for species diversity and aboveground biomass in semi-steppe rangelands. *Ecological Indicators* 93, 256-266

Comment 4:

Data preparation: For the time series with a higher than monthly resolution data are averaged (also more than one depth). I do not understand what is the purpose of this operation and I think it introduces a significant bias. Time series are not compared to each other, so I do not see the objective of the average. By doing that in some of the time series the analysis corresponds to a very specific observation whereas in others is an average of conditions. Considering the time scale of phytoplankton growth, in one you will have peaks because sampling coincided with a specific event, whereas in the other time series you will never get such a situation because you are averaging all the month conditions and smoothing all the peaks.

Response 4:

The main reason of using monthly averaged data throughout the analysis is that the results (i.e., system dynamics) based on state space reconstruction (e.g., CCM) are scale-dependent. The scale-dependency of state space reconstruction (SSR) has been widely recognized (e.g., Ref_X6-7: Gibson et al. 1992, *Physica D* & Dixon et al. 1999, *Science*). In fact, in dynamical systems, the dynamics are scale-dependent. That is, the driving factor that dominates at a monthly scale is not necessarily the factor that dominates at other scales (e.g., daily or annual) when applying CCM. Therefore, SSR-based methods require data preparation that standardizes time intervals among

data sets, at the same time scale that determines the scale of inferences. Such scale dependency is not a trivial assumption. For instance, the impacts of strong wind can be very influential to water column stability at a daily scale but might not be the main driving force at annual scales in which changes in water temperature may be the main driver. In addition, our collected datasets were from various aquatic systems with a variety of sampling resolutions, some that changed within a system. In our case, a monthly scale was the only consensus we could have for all datasets, and the monthly average was the most representative measure that we can use at this scale. Because we aim to carry out cross-system comparative analyses, a unified temporal scale is needed for studying dynamical systems. To clarify, we caveat our limitations in identifying the causal drivers at other temporal scales and advocated further investigations based on various temporal resolutions at **P17:L15-20**, “Lastly, due to limitations of data availability, our analysis only quantified causal strength across systems at a consensus monthly scale, acknowledging that state space reconstruction methods (e.g., CCM) are scale-dependent⁵⁵, e.g., one causal driver dominated monthly might not necessarily dominate at other time scales. Therefore, exploring causal feedbacks at other time scales needs further investigations by including more datasets with high temporal resolution and long-duration monitoring.”

We also clarify this issue and caveat potential problems caused by averaging in Method at **P20:L21-P21:L5**, “For consistency, monthly time series were generated by averaging over observations if sampling occurred on a finer timescale. Although such compilation potentially causes some inconsistency in smoothing temporal fluctuations of time series data among systems with various sampling frequencies, it was necessary because our methods based on state space reconstruction require time series data at equal intervals, dictating the temporal scale of analysis. In our case, monthly resolution is the only consensus that can be applied to all time series datasets and monthly average is the most representative measure at this scale.”

Nonetheless, we still carefully tackle the issues caused by differences among systems. Specifically, we standardized causal strengths and conducted cross-system comparisons among causal networks in a relative sense. We addressed this issue at **P24:L13-20**, “To address systematic differences in cross-map skill among study sites, which may arise due to differences in noise or time series length, causal strength⁹ was standardized. The standardized linkage strength (SLS) was calculated by dividing linkage strength (LS) by the maximum within each system: $SLS=LS/\max(LS)$; thus, SLS varied between 0 and 1 and indicated the relative importance with respect to the strongest causal link within the system. It is noteworthy that causal networks were

constructed and standardized separately for each system; i.e., it was not assumed that each system had equivalent dynamics and belonged to the same attractor.”

Ref_X6& Ref_55: Gibson, J. F., Doyne Farmer, J., Casdagli, M. & Eubank, S. An analytic approach to practical state space reconstruction. *Physica D: Nonlinear Phenomena* 57, 1-30 (1992).

Ref_X7: Dixon, P. A., Milicich, M. J., & Sugihara, G. Episodic fluctuations in larval supply. *Science*, 283, 1528-1530 (1999).

Comment 5:

Environmental factors: The objective of the paper is to modulate the effect of environmental factors and diversity on production. However, the environmental factors considered are limited to nutrients and temperature. If it is production other basic environmental factors such as irradiance, euphotic depth and water column stability should be included in the analysis. Otherwise we are excluding some of the main drivers of production from the analysis, which affects conclusions.

Response 5:

We did our best to collect data on additional factors from numerous systems as suggested by the reviewer, including irradiance, water column stability, and euphotic depth. However, these datasets were not available for all systems as now summarized in Table S3. Firstly, we considered irradiance and water column stability as exogenous environmental factors because they are unlikely to receive feedbacks from phytoplankton communities at the scale we studied. Water column stability was determined by maximal Brunt–Väisälä frequency in the water column and can be calculated from temperature vertical profiles without detailed hypsometry information; irradiance data were obtained from the weather stations or buoy stations near sampling sites. In total, we obtained long-term water column stability and irradiance datasets for 11 and 8 systems, respectively. Then we quantified causal effects of water column stability and irradiance on phytoplankton biomass and species richness. The quantified causal strengths on average were not as strong as the effects of diversity and nutrients. It is noteworthy that these findings were based on incomplete datasets and thus difficult to compare with causal strengths of the other factors based on all 19 datasets. Therefore, we considered these findings inconclusive, needing more detailed investigation after including more long-term datasets.

The aim of this study was not to heuristically test all possible causal factors; rather, we aimed to provide a novel methodological framework that can be applied to investigate causal feedbacks in natural systems. Therefore, we only presented these

results in Supplemental materials Fig. S4. and addressed our findings in the main text, **P10:L3:6**, “In addition to nutrients and temperature, effects of other environmental factors, such as irradiance and water column stability, were also presented in Fig. S4; however, these findings remained inconclusive, due to a lack of complete datasets across all systems (Table S3).”

For euphotic depth, this variable can be accessed in most systems through water transparency measures [e.g., Secchi depth or transparency measures from transparency meter (only in Lake Inba systems)], except for Western English Channel (Wc). However, the temporal variations of euphotic depth were largely determined by changing total phytoplankton biomass. Indeed, Chl a concentration had a strong negative relationship with euphotic depth in most systems (Fig. X4). This was not surprising, as the concentration of phytoplankton cells contributes the majority of suspended particles and thus is a direct cause of water transparency and euphotic depth, as algal cells physically absorbed irradiance in water, as suggested (e.g., Ref_X8: Tilzer 1998, Hydrobiologia). Although water transparency seems to affect phytoplankton growth, it is a composite factor implicitly summarizing influences of various limiting factors that cause self-regulation of phytoplankton communities. Therefore, we suggested that euphotic depth cannot be used effectively as an independent environmental factor; rather, it represented implicitly an influence of phytoplankton biomass. Therefore, we did not include euphotic depth analysis to the main results (Fig. 2 and S4) but only presented it in Fig. X4 in this response letter.

Ref_X8: Tilzer, M. M. Secchi disk—chlorophyll relationships in a lake with highly variable phytoplankton biomass. *Hydrobiologia*, 162, 163-171 (1988).

Fig. X4 Statistical relationships between chlorophyll a concentration and euphotic depth. The euphotic depth is determined from Secchi depth by multiplying a conversion factor (2.7) or from transparency measures determined by transparency meters (Lake Inba systems). Both datasets were log-transformed and normalized for each system. In all systems, chlorophyll a concentration was highly correlated with the approximated euphotic depth (mean correlation coefficient among systems: -0.562, $p < 0.001$ for all systems except for Ft ($p = 0.056$)).

Comment 6:

Overall, I would say this is a very strong paper on the methodological side, but the conclusions cannot be granted due to the extreme assumptions taken with the variables.

Response 6:

Following the reviewer's comments, we avoid stretching the assumptions made regarding the variables, as explained above. Consequently, the revised manuscript focuses more on presenting a novel methodological framework. We appreciate these comments from Reviewer #2, that strengthen our conclusions and make this study more focused.

Reviewer #3 (Remarks to the Author):

The article main claim is well presented in the abstract and title. It is that, concerning phytoplankton in aquatic bodies, the link between diversity and functioning, here primary production, is modulated by the environmental context. I found the demonstration of this claim outstanding. This demonstration is based on an exceptional data set of long time series in 19 aquatic systems. The analysis is then based on a numerical approach combining a perfectly complementary set of data analyses, combining pretty recently developed ones with more classical ones. Such results are of great interest in the community of ecology, with a wider possible impact more globally in biology, as it concerns a nested set of levels of organization, from the phytoplankton cell to the whole ecosystem, as the primary production is highly determinant for the whole functioning of the food web and other networks of interactions connected to the food web (forming what we call now a multiplex). An important output of this work will be to give the readers a new look on former Biodiversity / Functioning studies. By demonstrating that looking for generalities may not be our main objective in ecological studies, the authors should reorient the future protocols of data analysis in this domain. Another consequence of the authors' conclusion is to find a nice explanation of the apparent contradiction found previously in papers attending to join biodiversity and functioning, especially in the domain of plankton studies.

Response:

We appreciate this perfect summary of our work from Reviewer #3, pointing out the importance of our work in community ecology and in explaining contradictory results previously reported.

Comment 1:

Concerning the statistics chosen, I consider that the choices made in the set of statistical analyses and associated parameters is perfectly appropriate. I just have a few comments, presented hereafter, that may help improving the readability of their presentation.

1) Better explain within the text the difference in considering endogenous vs exogenous constraints and implications of such choices in terms of feedback loops. In fact, these explanation are very good in the figure 1 legend, but I think it is missing from the text of the introduction.

Response 1:

Thank you for pointing out the appropriateness of the statistical methods used in this study. Following the reviewer's suggestion, we revised the Introduction to include

more detailed explanations considering endogenous and exogenous environmental factors at **P7:L9-12**, “However in natural systems, debate remains over whether the effect of diversity on ecosystem functioning is stronger than exogenous (L2 in Figure 1) or endogenous drivers²⁵ (L8 in Figure 1), both of which affect organisms⁵, although only endogenous drivers can be affected by organisms and involved in feedbacks²⁶.”

Ref_5: Miki, T., Ushio, M., Fukui, S. & Kondoh, M. Functional diversity of microbial decomposers facilitates plant coexistence in a plant–microbe–soil feedback model. *Proc. Natl. Acad. Sci. U.S.A.* 107, 14251-14256 (2010).

Ref_25: Mittelbach, G. G. et al. What is the observed relationship between species richness and productivity? *Ecology* 82, 2381-2396 (2001).

Ref_26: Enoki, T. et al. Progress in the 21st century: a roadmap for the Ecological Society of Japan. *Ecological Research* 29, 357-368 (2014).

Comment 2:

2) Better explain the way the phytoplankton species richness is determined, if there was any process of quality control, in order to ensure the comparability of the data sets. I also missed the information about how the level of the species determination was reached, e.g. was electronic microscopy added, when needed, to optic microscopy.

Response 2:

In all datasets, composition data had species-level taxonomic resolution based on microscopic enumeration, and species richness was defined as the number of species present at each site, on each sampling date. It may be an exaggeration to claim that all the phytoplankton composition data followed exactly the same methods in a large, compiled dataset across various systems. However, the methods used to derive the composition data were similar. In the revision, we included more detail on counting methods listed in Table S2 and clarified our methods for the investigation of phytoplankton composition and species richness at **P20:L10-15**, “Phytoplankton samples were identified to the finest taxonomical level (generally species level if possible) and enumerated under an optical microscope, based on counting methods summarized in Table S2. The counting methods used were similar (e.g., Utermöhl⁶³ method and relevant approaches). Based on composition data, species richness was derived and defined as the number of species present in the phytoplankton community.”

Nonetheless, we still carefully tackle the issues caused by differences among systems. Specifically, we standardized causal strengths and conducted cross-system

comparisons among causal networks in a relative sense. We addressed this issue at **P24:L13-20**, “To address systematic differences in cross-map skill among study sites, which may arise due to differences in noise or time series length, causal strength⁹ was standardized. The standardized linkage strength (SLS) was calculated by dividing linkage strength (LS) by the maximum within each system: $SLS=LS/\max(LS)$; thus, SLS varied between 0 and 1 and indicated the relative importance with respect to the strongest causal link within the system. It is noteworthy that causal networks were constructed and standardized separately for each system; i.e., it was not assumed that each system had equivalent dynamics and belonged to the same attractor.”

Ref_9: Chang, C.-W. et al. Long-term warming destabilizes aquatic ecosystems through weakening biodiversity-mediated causal networks. *Global Change Biology* 26, 6413-6423 (2020).

Ref_63: Utermöhl, H. Zur Vervollkommnung der quantitativen Phytoplankton-Methodik: Mit 1 Tabelle und 15 abbildungen im Text und auf 1 Tafel. *Internationale Vereinigung für theoretische und angewandte Limnologie: Mitteilungen* 9, 1-38 (1958).

Comment 3:

3) Please explain further how chl a concentrations were translated into production values.

Response 3:

Following the comments from Reviewer #2, we used chlorophyll *a* concentration (Chl*a*) as a proxy for phytoplankton biomass instead of a proxy for productivity. Thus, we replaced all the wording of ‘production’ by ‘biomass.’ In addition, we provided statistical justification for the use of Chl*a* as a good proxy for phytoplankton community biomass in Fig. X3 below. There were strong positive relationships between Chl*a* and phytoplankton community biomass derived from composition datasets (total sum of individual species biomass). We avoided using total biomass data, derived from our composition dataset, because this measure is statistically dependent on our diversity measures (also derived from composition data). In addition, the use of Chl*a* as a proxy of algal biomass is widely applied in the BDEF literature (Ref_17 & 60). Therefore, we only presented Fig. X3 in the response letter and addressed the use of Chl*a* as a proxy of phytoplankton biomass in Methods at **P20:L7-9**, “(2) chlorophyll-a concentration as a measure of phytoplankton biomass and ecosystem function, a widely used proxy of algal biomass in the BDEF literature^{17,62}.”

Ref_17: Cardinale, B. J. Biodiversity improves water quality through niche partitioning. *Nature* 472, 86-89, (2011).

Ref_62: Lewandowska, A. M. et al. The influence of balanced and imbalanced resource supply on biodiversity–functioning relationship across ecosystems. *Philosophical Transactions of the Royal Society B: Biological Sciences* 371, (2016).

Fig. X3 Statistical relationships between chlorophyll a concentration and community biomass derived from composition data (both were log transformed and normalized) for each system. In most systems, chlorophyll a concentration was highly correlated with community biomass derived from composition data (mean correlation coefficient= 0.668, $p < 0.001$ for all systems except for Ft ($p = 0.76$)), except for Feitsui Reservoir (Ft) in which small phytoplankton cells were not effectively collected by a 10-um plankton net.

Comment 4:

4) I also missed information on the choice of CCA rather than RDA in the multivariate analyses. The connection between explanatory and explained variables would not follow the same model in the two cases, and the choice of CCA is probably justified, but the confirmation should be given in the text. What is the justification of the choice of a modal rather than linear model in the relationship? Did you use coenoclines to drive this choice?

Response 4:

Following the reviewer’s comments, we justified the use of direct gradient analysis according to the procedures previously suggested (Šmilauer and Lepš 2014, Cambridge university press). Firstly, we performed preliminary detrended correspondence analysis (DCA) on the dataset comprised of linkage strengths or loop

weights. Then we examined the coenoclines that depicted the quantitative relationships between various state variables and the gradient represented by the first axis of the DCA (Fig. S10). Based on coenoclines, we identified some unimodal relationships. However, due to the short length of gradients, there were more linear than unimodal relationships. According to the criterion proposed in Šmilauer and Lepš (2014), RDA outperforms CCA when gradient length is < 3 . Thus, we changed all direct gradient analyses presented in this study from CCA to RDA analysis. Nonetheless, the ordination results based on RDA were qualitatively the same with the original results based on CCA, with very minor improvements in explained variation (0.1-2.9%). Therefore, all our conclusions remained the same and are robust to the use of various direct gradient analyses. The justifications of direct gradient analysis were provided in Supplemental Materials (Fig. S10) and addressed in Methods at **P25:L18-20**, “The use of RDA instead of CCA was justified based on our analysis on coenoclines (Fig. S10), with more linear coenoclines and a short gradient length (less than 3)⁷⁹.”

Ref_79: Šmilauer, P. & Lepš, J. Multivariate analysis of ecological data using CANOCO 5. (Cambridge university press, 2014).

Comment 5:

5) Please give the number of iterations of the permutation tests.

Response 5:

We used $n=10000$ as the number of iterations. This information was included in Methods at **P25:L20-22**, “All permutation tests performed in this study were based on the null distribution generated from 10000 random permutations.”

Thank you very much for these comments that certainly improved the clarity of our manuscript and strengthened our multivariate analyses.

Reviewers' Comments:

Reviewer #1:

Remarks to the Author:

The authors use empirical dynamic modeling to (EDM) study the interactions of biodiversity and ecosystem function (total biomass) of phytoplankton communities in 19 lakes, globally spread, for which long time series are available. Using the EDM approach, they found nonlinear feedback effects between biodiversity and biomass, and between these two properties and nutrients N and P. In particular, the authors found that biodiversity was more important than effects from the external environment as a driver of biomass. They also found important three-variable feedback loops for, biodiversity-biomass-N and biodiversity-biomass-P in warm and cold temperature lakes, respectively. These are new results for macroecology showing the importance of internal feedbacks in phytoplankton.

The very detailed responses of the authors help clarify the questions that I had about the original version of this paper, and I the questions of the other reviewers. It is encouraging to see the EDM approach applied to the biodiversity-ecosystem function question, which is one of the most important ecological issues and difficult to resolve. The equation-free modeling does not provide mechanistic explanations for these results, but it should stimulate experiments and modeling. As the authors note in their rebuttal, lake models such as BLOOM do not contain enough phytoplankton diversity to represent phytoplankton biomass. It might be possible in the future to reach levels of 15.1 - 38.9 species in lake models that the authors mention as being necessary, though perhaps that is not easy. Existing lake models, such as PCLake, show that stratification status has an effect of Chl-a, etc. on phytoplankton biomass, so more information on such lake characteristics would be useful to put into further studies EDM when it becomes possible.

I found only one grammar issue. On page 18, line 9, change 'explore more detail causal feedbacks' to 'explore in more detail causal feedbacks'.

Reviewer #2:

Remarks to the Author:

The authors have made a great effort to address the comments but the main issue remains, the basic data do not allow to address the scientific questions the authors want to answer. This is acknowledged by the authors in the answer to the review:

"The aim of this study was not to heuristically test all possible causal factors; rather, we aimed to provide a novel methodological framework that can be applied to investigate causal feedbacks in natural systems."

But this acknowledgment remain in contrast with the the objectives staed in the project:

"we aimed to understand biodiversity in aquatic ecosystems by addressing the following questions: 1) Under what conditions are phytoplankton diversity effects on ecosystem functioning stronger than the effects of environmental drivers? 2) What is the strongest causal determinant for species diversity? 3) What are the most effective pathways through which changes in diversity 182 propagate to other parts of the network, and feedback on themselves? 4) Are there any emerging macroecological patterns explaining how causal links, pairwise feedbacks, and triangular feedbacks vary along large-scale 185 environmental gradients?"

To explicitly answer these questions, we performed cross-system comparisons on the reconstructed causal networks to evaluate: (i) the relative importance among causal links affecting phytoplankton biomass; (ii) the relative importance among causal links affecting phytoplankton diversity; (iii) the relative strengths of more complex feedbacks involving biodiversity; and iv) how the strengths of the network modules investigated in (i)-(iii) vary with environmental characteristics."

Some of the issues cannot be solved, the dispersion on the analysis the authors have carried out on the relation between chl a and biomass confirms my opinion that Chl a cannot be used as

biomass proxy to answer questions about subtle effects. The presented figure X3 in the response (in log log) shows that for the same Chl a the range of real biomasses is huge. Clearly due to factors not considered such as irradiance, light attenuation etc. Therefore the data confirm my opinion that Chl a cannot be used in this context. Obviously, as the authors indicate, using the counts (if they have volume estimates for the different species) to analyse biomass effects on diversity is tricky because the same data. But still better than using chl a.

Basically the answer of the authors confirms my previous opinion. Methodologically, is really an excellent database, but the basic data do not allow to answer the questions the objectives addressed.

Reviewer #3:

Remarks to the Author:

The manuscript can be accepted in its present form. The authors perfectly took into account all my comments and modified the statistical analyses and the text accordingly.

Reviewer #4:

Remarks to the Author:

Review Chang et al.

The paper by Chang et al. uses convergent cross mapping to assess, rank and disentangle causal relationships between phytoplankton species richness, standing stock of plankton as indicated by chlorophyll-a biomass, as well as a range of exogenous and endogenous environmental factors such as nutrient concentrations and surface temperature for 19 aquatic and primarily limnic ecosystems.

I was asked to consider issues raised by reviewer 2 in the first round of review with regard to the use of chlorophyll-a as a proxy for phytoplankton biomass and/or productivity.

Despite the fact that chlorophyll-a is a very frequently used, very common proxy for phytoplankton biomass both in marine as well as in lake ecosystems, I concur with reviewer 2 that the use of a biomass proxy which blends environmental factors (irradiation, self-shading, mixing regime) with biotic ones into one variable such as cell density needs to be critically evaluated in a study that tries to rank very subtle and usually weak effects between ecosystem constituents, ecosystem functions and the physico-chemical environment. I am also not fully satisfied by the response of the authors to this important point. In their response, the authors show a log-log biomass correlation plot between chlorophyll-a based biomass and count-based phytoplankton biomass, which shows a strong correlation of $\rho=0.668$ between both measures, albeit in log-log space. Yet the plot also shows a substantial range of variability in terms of the slope of the relationships, the residuals, and also differences between systems. Considering the substantial amount of unexplained variance, and the fact that weak relationships are being explored with this novel methodology, I think this issue needs to be treated in the main text of the manuscript, and also mentioned in the caveats section of the paper. The authors do not explain in their response to reviewer 2's initial comment how the biomass of the system was calculated based on the plankton counts – if species-specific abundance-biomass conversion factors were used, this would have been a major endeavor, and the conversion alone would lead to a huge range of uncertainty, which would explain some of the biomass-biomass misfit. The authors then continue to argue that they did not use these in situ biomass estimates due to statistical correlations with their diversity metrics. While I understand that there may be a correlation between the Shannon index and their in situ cell count based biomass estimates, essentially, I don't understand why this should be the case for their species richness estimates. Typically, phytoplankton blooms are dominated by few species with high biomass, with rapid species succession patterns of different species throughout the duration of a bloom, and I do not see why there should be a correlation between the biomass and richness, nor why this should be a problem, since everything else they use is also correlated. NO₃ concentrations tends to be highly correlated with chlorophyll-a, too, at least at regional to

global scales. In my view, to assess causal relationships the authors should actually base their network analysis on exactly this additional biological data (abundances of exactly those species that add to the diversity estimates determined experimentally) they have – and thus should use the biomasses they have to demonstrate that the use of chlorophyll-a as a proxy is indeed justified, and leads to the same conclusions. This is also because the data is based on observations using a specific mesh size to collect phytoplankton species that is prone to missing a range of smaller organisms (nano/picophytoplankton), yet total chlorophyll-a will include that of the smaller species whose diversity (a) we now know based on metagenomic studies is large, and (b) whom we cannot identify morphologically using traditional methods. Their analysis should be repeated based on the total abundance (or biomass) estimates they have, and compared qualitatively and quantitatively to the results obtained with chlorophyll-a as a proxy in the main text. Furthermore, satellite remote sensing information paired with in situ chlorophyll-a measurements could be used to assess causal links with productivity. This would build trust that the conclusions obtained are robust.

I further share reviewer 2's concerns about the use of a monthly integration time scale to assess causal links in phytoplankton systems when phytoplankton cells tend to have turnover times of 1-2 days. At temperate and high latitudes, blooms last weeks, not months, and species composition (and species richness) varies strongly throughout each bloom. Here again, I am not fully convinced by the authors' response with regard to this issue, and I would like them to repeat their analysis for each time series at the time scales that measurements are taken. This would allow for a comparison of the drivers and feedback strengths across the temporal scales available, and would allow for the quantification of the effect of temporal averaging.

My main concern, however, is that zooplankton are not included in this analysis. This is mentioned briefly in the caveats, but not in the detail I think is warranted here. In my view, this is a major concern and one of the key issues that may render some of the conclusions invalid. Zooplankton have long been monitored in lakes, and again I do not buy the explanation that these counts have been unavailable. In some of the 19 lakes perhaps, but we do not need 19 lakes to evaluate – and quantify – the effect of top-down pressures. Zooplankton grazing has been shown to be one of the primary drivers of phytoplankton diversity in many modeling studies, and top-down factors have also been identified a key determinant of lower trophic level network structure in recent global metagenomic studies. Due to the inverse biomass pyramid in marine and many freshwater systems, top-down pressure are much more important in aquatic systems than in terrestrial ones, and invasive species, such as the jellyfish *M. leidyi* have shown to change the phenological patterns of entire inland seas, such as the Caspian Sea. Hence, if there are causal links between diversity and biomass and nutrients, then those must not neglect the primary cause of phytoplankton demise, which is grazing by zooplankton. Zooplankton both store and integrate nutrient signals contained in the biomass of primary producers, and have also been found, and been suggested to control chlorophyll-a concentration in natural and modeled systems.

The CCM method might be new, and attractive for the conceptual study of causal links in aquatic systems, but the authors also write that it is robust to moderate noise – whereas the noise in aquatic systems is high. Furthermore, any statistical algorithm will have difficulties in separating the influences of several highly correlated variables – in most marine systems, pretty much everything, from temperature to nutrient dynamics to mixing is correlated at larger spatio-temporal scales. Hence, great care needs to be taken to avoid the identification of spurious links or methodological artifacts as patterns. The authors have thrown a powerful new method at their causality problem, but currently still fail to document their assumptions, and in particular the limits and limitations of their method in detail. What is the effect of each methodological choice taken, what is the uncertainty of this approach? While I am very excited about the approach, and think that this methodology has a lot of potential, I still fail to be convinced yet that this is not an example of a 'garbage in – garbage out' model. To convince me, I would need to see a more careful assessment of the method, variable selection (primary producer richness, standing stock, nutrients, temperature), the choice of proxies, etc. A sound sensitivity analysis, and solid uncertainty estimates where critical choices are being made would certainly convince me - and perhaps also reviewer 2.

To conclude, I do agree with reviewer 2 that a second round of review is warranted, but I don't

agree that 'the data does not allow to answer the questions addressed'. This work has potential, and the method is exciting, and I would love to be convinced that what we see are actual signals.

Comments from reviewers are copied below with our point-by-point responses. We use black text for the reviewer's comments and blue text for our responses. Changes in the text of the revised manuscript are underlined.

Reviewer #1 (Remarks to the Author):

The authors use empirical dynamic modeling to (EDM) study the interactions of biodiversity and ecosystem function (total biomass) of phytoplankton communities in 19 lakes, globally spread, for which long time series are available. Using the EDM approach, they found nonlinear feedback effects between biodiversity and biomass, and between these two properties and nutrients N and P. In particular, the authors found that biodiversity was more important than effects from the external environment as a driver of biomass. They also found important three-variable feedback loops for, biodiversity-biomass-N and biodiversity-biomass-P in warm and cold temperature lakes, respectively. These are new results for macroecology showing the importance of internal feedbacks in phytoplankton.

The very detailed responses of the authors help clarify the questions that I had about the original version of this paper, and I the questions of the other reviewers. It is encouraging to see the EDM approach applied to the biodiversity-ecosystem function question, which is one of the most important ecological issues and difficult to resolve. The equation-free modeling does not provide mechanistic explanations for these results, but it should stimulate experiments and modeling. As the authors note in their rebuttal, lake models such as BLOOM do not contain enough phytoplankton diversity to represent phytoplankton biomass. It might be possible in the future to reach levels of 15.1 - 38.9 species in lake models that the authors mention as being necessary, though perhaps that is not easy. Existing lake models, such as PCLake, show that stratification status has an effect of Chl-a, etc. on phytoplankton biomass, so more information on such lake characteristics would be useful to put into further studies EDM when it becomes possible.

Response:

Thank you for this excellent summary of our work. We are also grateful for the insightful comments made by the Reviewer. Indeed, applying EDM is a beginning for studying those key ecological issues that are difficult to investigate at large spatiotemporal scales, such as biodiversity-ecosystem function in aquatic systems. Our works provide a useful methodological framework that reveals the importance of various causal links and feedbacks and presents novel macroecological patterns indicating how their importance varied along environmental gradients. Nonetheless,

we totally agree with the Reviewer that it would require more sophisticated experiments and models to fully solve the underlying mechanisms and eventually quantitatively forecast the consequences of biodiversity loss.

Comment:

I found only one grammar issue. On page 18, line 9, change 'explore more detail causal feedbacks' to 'explore in more detail causal feedbacks'.

Response:

We corrected the grammar issue according to the Reviewer's suggestion.

Reviewer #2 (Remarks to the Author):

The authors have made a great effort to address the comments but the main issue remains, the basic data do not allow to address the scientific questions the authors want to answer. This is acknowledged by the authors in the answer to the review:

"The aim of this study was not to heuristically test all possible causal factors; rather, we aimed to provide a novel methodological framework that can be applied to investigate causal feedbacks in natural systems."

But this acknowledgment remain in contrast with the the objectives staed in the project:

"we aimed to understand biodiversity in aquatic ecosystems by addressing the following questions: 1) Under what conditions are phytoplankton diversity effects on ecosystem functioning stronger than the effects of environmental drivers? 2) What is the strongest causal determinant for species diversity? 3) What are the most effective pathways through which changes in diversity propagate to other parts of the network, and feedback on themselves? 4) Are there any emerging macroecological patterns explaining how causal links, pairwise feedbacks, and triangular feedbacks vary along large-scale environmental gradients?"

To explicitly answer these questions, we performed cross-system comparisons on the reconstructed causal networks to evaluate: (i) the relative importance among causal links affecting phytoplankton biomass; (ii) the relative importance among causal links affecting phytoplankton diversity; (iii) the relative strengths of more complex feedbacks involving biodiversity; and iv) how the strengths of the network modules investigated in (i)-(iii) vary with environmental characteristics."

Some of the issues cannot be solved, the dispersion on the analysis the authors have carried out on the relation between chl a and biomass confirms my opinion that Chl a cannot be used as biomass proxy to answer questions about subtle effects. The presented figure X3 in the response (in log log) shows that for the same Chl a the range of real biomasses is huge. Clearly due to factors not considered such as irradiance, light attenuation etc. Therefore the data confirm my opinion that Chl a cannot be used in this context. Obviously, as the authors indicate, using the counts (if they have volume estimates for the different species) to analyse biomass effects on diversity is tricky because the same data. But still better than using chl a.

Basically the answer of the authors confirms my previous opinion. Methodologically, is really an excellent database, but the basic data do not allow to answer the questions the objectives addressed.

Response:

In our previous response, we suggested that the diversity and biomass data derived from composition data were statistically very correlated. However, as pointed out by Reviewer #4, this is true only when using abundance-based diversity indices, such as Shannon diversity, but may be appropriate when considering species richness as the diversity index. Therefore, we conducted CCM analysis based on the composition converted biomass data. In addition to the statistical relationships between *Chla* and composition converted biomass (Fig. S10a-b), causal strengths as well as loop weights that are associated with composition converted biomass were highly correlated with the strengths derived from analysing *Chla* (Fig. S10c-d; Pearson $r=.626$ and $.665$, respectively). Consequently, our main conclusions about the relative importance among various causal links and feedbacks (Fig. S11a-d; an analogue of Fig. 2) and their associations with environmental factors (Fig. S11e-h; an analogue of Fig. 4) were qualitatively very similar with findings based on *Chla*. For example, species richness still importantly affected composition converted biomass compared to the other environmental factors, such as nutrients, irradiance, and water stability (Fig. S11a). In addition, the RDA ordinations (Fig.S11e-h) based on the same sets of environmental variables manifested very similar patterns to those presented in Figure 4. However, these ordinations explained less variation compared to the ordinations based on *Chla* [proportions of explained variations (RDA 1+RDA 2) = 33.1%, 33.6%, 41.8%, and 49.4% were less than those presented in Fig. 4 =55.9%, 40.0%, 46.7%, and 57.3%, respectively]. Nonetheless, these findings indicated our main conclusions are robust to the use of alternative biomass measure.

Figure S10 | Quantification of causal strength was robust to the use of the alternative biomass measure converted from phytoplankton composition data.

Figure S11 | The findings of main analyses were robust to the use of the alternative biomass measure converted from phytoplankton composition data.

The most precise way to determine phytoplankton biomass is probably by measuring the dry weight or carbon content of phytoplankton cells isolated from water samples. Unfortunately, this approach is technically difficult to implement in long-term monitoring. Consequently, phytoplankton biomass was often inferred from either Chl a concentration or composition data in previous literatures. Even if these indices are rough indicators of phytoplankton biomass, our results based on the CCM analysis of Chl a revealed the important role of phytoplankton diversity in driving long-term Chl a dynamics. Nevertheless, it remains a possibility that applying CCM enables to quantify the causal links associated with alternative biomass measures as shown in Figs. S10-11. Moreover, we suggest that the biomass measure converted from composition data is not always superior to the proxy of Chl a , especially in cross-system analyses. As pointed out by Reviewer #4, measures converted from composition data include substantial uncertainty; this likely contributes to the deviations presented in Fig. S10a-b. This is because converting cell counts to biomass depends on species-specific abundance-biomass conversion factors that are often assumed to be fixed values in the absence of detailed measurements of individual cell size and differ among systems. This inconsistency might partly explain why variations in causal strengths and feedbacks based upon biomass conversion data were less well explained by cross-system environmental gradients than those derived from Chl a measured by a standard chemical approach across systems. Although Chl a has slightly different meanings compared to true phytoplankton biomass, it is still a useful functional index inferring phytoplankton stock with more emphasis on the capacity of phytoplankton for photosynthesis in various types of lakes. Presenting Chl a as an effective index indicating *ecosystem functioning* in aquatic systems, our aim for providing a methodological framework that evaluates the causal links and feedbacks among diversity, *ecosystem functioning*, and environmental factors can be achieved. For instance, Chl a was often used as a functional index inferring aquatic ecosystem functioning in numerous biodiversity-ecosystem functioning studies (e.g., RefX1-3).

RefX1: Gerhard, M., C. Mori and M. Striebel. (2021) Nonrandom species loss in phytoplankton communities and its effect on ecosystem functioning. *Limnology and Oceanography* 66:779-792.

RefX2: Duffy, J. E., C. M. Godwin and B. J. Cardinale. (2017) Biodiversity effects in the wild are common and as strong as key drivers of productivity. *Nature* 549:261-264.

RefX3: Cardinale, B. J. (2011) Biodiversity improves water quality through niche partitioning. *Nature* 472:86-89.

Therefore, we included the analysis of composition converted biomass in the Supplementary Information (Figs. S10-11) and caveated the use of alternative biomass measures in aquatic systems at P17:L13- P18:L9, “

Sensitivity analysis using composition converted biomass measure

Our conclusions were robust to the use of alternative biomass measure. Specifically, the main findings (Figs. 2 and 4) based on the analysis of phytoplankton biomass inferred by Chl a were qualitatively similar with findings based on composition converted biomass (Figs. S10-11). Although the relationship between Chl a and true phytoplankton biomass varies with environmental conditions (e.g., light), it remains an effective functional index inferring phytoplankton stock with more emphasis on photosynthesis capacity (i.e., biomass of photosynthetic machines). In contrast, composition converted biomass, though has similar meaning with overall phytoplankton biomass, contains high uncertainty by assuming species-specific conversion factors, especially when measurement of individual cells size was lacking. In addition, these conversion factors were determined by various geometrical models, which differ among systems; this is in contrast to the standard chemical approach used in determining Chl a , which makes Chl a more suitable to reveal cross-system variations in causal strengths along environmental gradients (Fig. 4 and Fig. S11e-h). However, Chl a integrates all kinds of photoautotrophs that might not be fully included in counting data (e.g., picoplankton). Thus, investigating diversity effects of more complete phytoplankton groups requires novel techniques (e.g., metagenomics⁵⁵). Nonetheless, it remains an open question about how the presented causal links and feedbacks change when considering various types of functional indices and diversity measures.”

Reviewer #3 (Remarks to the Author):

The manuscript can be accepted in its present form. The authors perfectly took into account all my comments and modified the statistical analyses and the text accordingly.

Response:

Thank you very much for your constructive comments that substantially improve our publication.

Reviewer #4 (Remarks to the Author):

Review Chang et al.

The paper by Chang et al. uses convergent cross mapping to assess, rank and disentangle causal relationships between phytoplankton species richness, standing stock of plankton as indicated by chlorophyll-a biomass, as well as a range of exogenous and endogenous environmental factors such as nutrient concentrations and surface temperature for 19 aquatic and primarily limnic ecosystems.

I was asked to consider issues raised by reviewer 2 in the first round of review with regard to the use of chlorophyll-a as a proxy for phytoplankton biomass and/or productivity.

Despite the fact that chlorophyll-a is a very frequently used, very common proxy for phytoplankton biomass both in marine as well as in lake ecosystems, I concur with reviewer 2 that the use of a biomass proxy which blends environmental factors (irradiation, self-shading, mixing regime) with biotic ones into one variable such as cell density needs to be critically evaluated in a study that tries to rank very subtle and usually weak effects between ecosystem constituents, ecosystem functions and the physico-chemical environment. I am also not fully satisfied by the response of the authors to this important point. In their response, the authors show a log-log biomass correlation plot between chlorophyll-a based biomass and count-based phytoplankton biomass, which shows a strong correlation of $\rho=0.668$ between both measures, albeit in log-log space. Yet the plot also shows a substantial range of variability in terms of the slope of the relationships, the residuals, and also differences between systems. Considering the substantial amount of unexplained variance, and the fact that weak relationships are being explored with this novel methodology, I think this issue needs to be treated in the main text of the manuscript, and also mentioned in the caveats section of the paper. The authors do not explain in their response to reviewer 2's initial comment how the biomass of the system was calculated based on the plankton counts – if species-specific abundance-biomass conversion factors were used, this would have been a major endeavor, and the conversion alone would lead to a huge range of uncertainty, which would explain some of the biomass-biomass misfit. The authors then continue to argue that they did not use these in situ biomass estimates due to statistical correlations with their diversity metrics. While I understand that there may be a correlation between the Shannon index and their in situ cell count based biomass estimates, essentially, I don't understand why this should be

the case for their species richness estimates. Typically, phytoplankton blooms are dominated by few species with high biomass, with rapid species succession patterns of different species throughout the duration of a bloom, and I do not see why there should be a correlation between the biomass and richness, nor why this should be a problem, since everything else they use is also correlated. NO₃ concentrations tends to be highly correlated with chlorophyll-a, too, at least at regional to global scales. In my view, to assess causal relationships the authors should actually base their network analysis on exactly this additional biological data (abundances of exactly those species that add to the diversity estimates determined experimentally) they have – and thus should use the biomasses they have to demonstrate that the use of chlorophyll-a as a proxy is indeed justified, and leads to the same conclusions. This is also because the data is based on observations using a specific mesh size to collect phytoplankton species that is prone to missing a range of smaller organisms (nano/picophytoplankton), yet total chlorophyll-a will include that of the smaller species whose diversity (a) we now know based on metagenomic studies is large, and (b) whom we cannot identify morphologically using traditional methods. Their analysis should be repeated based on the total abundance (or biomass) estimates they have, and compared qualitatively and quantitatively to the results obtained with chlorophyll-a as a proxy in the main text. Furthermore, satellite remote sensing information paired with in situ chlorophyll-a measurements could be used to assess causal links with productivity. This would build trust that the conclusions obtained are robust.

Response:

We agree with the comments from Reviewer #4 that the statistical correlation is an issue only when applying abundance-based diversity indices, such as Shannon diversity, but may be less influential when considering species richness. Therefore, we followed the Reviewer's comment to conduct CCM analysis based on composition converted biomass data (log transformed) and then compared the results with our main findings based on the analysis of Chl_a. In addition to the statistical relationships between Chl_a and composition converted biomass (Fig. S10a-b), the causal strengths and loop weights that are associated with composition converted biomass were highly correlated with the corresponding strengths derived from Chl_a (Fig. S10c-d; Pearson $r=.626$ and $.665$, respectively). Consequently, our findings about the relative importance among various causal links and feedback (Fig. S11a-d; an analogue of Fig. 2) and their associations with environmental gradients (Fig. S11e-h; an analogue of Fig. 4) were qualitatively very similar to the main findings based on Chl_a. For example, species richness still importantly affected composition converted biomass

compared to the other environmental factors, such as nutrients, irradiance, and water stability (Fig. S11a). In addition, the RDA ordinations (Fig.S11e-h) based on the same sets of environmental variables manifested very similar patterns to those presented in Figure 4. However, these ordinations explained less variation compared to the ordinations based on *Chla* [proportion of explained variations (RDA 1+RDA 2) = 33.1, 33.6, 41.8, and 49.4% in Fig. S11e-h compared to the proportions obtained from analysing $Chla=55.9, 40.0, 46.7, \text{ and } 57.3\%$ in Fig. 4a-d, respectively], indicating a weaker association between these causal strengths and cross-system environmental gradients than the associations revealed by analysing *Chla*. Nonetheless, these findings still suggested the robustness of our main conclusions to the use of alternative biomass measure.

Figure S10 | Quantification of causal strength was robust to the use of the alternative biomass measure converted from phytoplankton composition data.

Figure S11 | The findings of main analyses were robust to the use of the alternative biomass measure converted from phytoplankton composition data.

As pointed out by the Reviewer, the composition converted biomass was mostly calculated from the counted cell numbers using species-specific abundance-biomass conversion factors (e.g., mean cell volumes calculated from various geometrical models), which potentially lead to a huge range of uncertainty. Although the use of composition converted biomass has some advantages on results interpretation, we suggested this biomass measure is not superior to the proxy of Chl a , especially in this cross-system analysis. This likely contributes to the deviations presented in Fig. S10a-b; this is because converting cell counts to biomass depends on species-specific abundance-biomass conversion factors that are often assumed to be fixed values in the absence of detailed measurements of individual cell size and differ in various systems. This inconsistency might partly explain why variations in causal strengths and feedbacks based upon biomass conversion data were less well explained by cross-system environmental gradients than those derived from Chl a measured by a standard chemical approach across systems. Although Chl a has slightly different meanings compared to true phytoplankton biomass, it is still a useful functional index inferring phytoplankton stock, with more emphasis on the capacity of phytoplankton for photosynthesis in various types of lakes. Presenting Chl a as an effective index indicating *ecosystem functioning* in aquatic systems, our aim for providing a methodological framework that evaluates the causal links and feedbacks among diversity, *ecosystem functioning*, and environmental factors can be achieved. For instance, Chl a was often used as a functional index inferring aquatic ecosystem functioning in numerous biodiversity-ecosystem functioning studies (e.g., RefX1-3).

RefX1: Gerhard, M., C. Mori and M. Striebel. (2021) Nonrandom species loss in phytoplankton communities and its effect on ecosystem functioning. *Limnology and Oceanography* 66:779-792.

RefX2: Duffy, J. E., C. M. Godwin and B. J. Cardinale. (2017) Biodiversity effects in the wild are common and as strong as key drivers of productivity. *Nature* 549:261-264.

RefX3: Cardinale, B. J. (2011) Biodiversity improves water quality through niche partitioning. *Nature* 472:86-89.

We also add in discussion a caveat to acknowledge that the phytoplankton diversity (as estimated from counting data) included only some phytoplankton groups but not all phytoplankton groups, whereas Chl a (here, a biomass proxy) data potentially include all photoautotrophs. Therefore, there is potentially a mismatch. Thus,

following the Reviewer's comment, we suggested future work using novel sampling techniques, e.g., metagenomics, that shall be helpful in revealing the diversity effects from all phytoplankton groups. In our revision, we included the analysis of composition converted biomass in the Supplementary Information (Fig. S10-11) and caveated the issues of alternative biomass measures in a new paragraph at P17:L13-P18:L9, “

Sensitivity analysis using composition converted biomass measure

Our conclusions were robust to the use of alternative biomass measure. Specifically, the main findings (Figs. 2 and 4) based on the analysis of phytoplankton biomass inferred by Chl_a were qualitatively similar with findings based on composition converted biomass (Figs. S10-11). Although the relationship between Chl_a and true phytoplankton biomass varies with environmental conditions (e.g., light), it remains an effective functional index inferring phytoplankton stock with more emphasis on photosynthesis capacity (i.e., biomass of photosynthetic machines). In contrast, composition converted biomass, though has similar meaning with overall phytoplankton biomass, contains high uncertainty by assuming species-specific conversion factors, especially when measurement of individual cells size was lacking. In addition, these conversion factors were determined by various geometrical models, which differ among systems; this is in contrast to the standard chemical approach used in determining Chl_a, which makes Chl_a more suitable to reveal cross-system variations in causal strengths along environmental gradients (Fig. 4 and Fig. S11e-h). However, Chl_a integrates all kinds of photoautotrophs that might not be fully included in counting data (e.g., picoplankton). Thus, investigating diversity effects of more complete phytoplankton groups requires novel techniques (e.g., metagenomics⁵⁵). Nonetheless, it remains an open question about how the presented causal links and feedbacks change when considering various types of functional indices and diversity measures.”

The Reviewer made an interesting suggestion about pairing satellite remote sensing with *in situ* Chl_a measurements. However, applications of satellite remote sensing fall shorts in identifying subsurface chlorophyll maximum, which often causes the underestimation of phytoplankton abundance (Ref X4). In addition, some of our studied sites were shallow where the sediment resuspension can easily interfere remote sensing observations at surface, unless thoughtful corrections can be applied (Ref X5). Furthermore, satellite remote sensing is critically affected by water turbidity and CDOM that vary among our studied systems, making it difficult to infer phytoplankton biomass in a consistent way. Moreover, satellite data for inferring water colour (as a proxy for phytoplankton biomass) are often missing due to weather

conditions (e.g., clouds). Considering those limitations, we did not include the analysis of remote sensing data in this revision. Employing remote sensing data to infer phytoplankton biomass requires careful ground truthing, which is beyond the scope of this study.

Ref X4: Kutser T. (2004) Quantitative detection of chlorophyll in cyanobacterial blooms by satellite remote sensing. *Limnology and Oceanography* 49(6):2179-2189.

Ref X5: Kutser T., J. Hedley, C. Giardino, C. Roelfsema and V. E. Brando. (2020) Remote sensing of shallow waters – A 50 year retrospective and future directions. *Remote Sensing of Environment* 240:111619.

Comments:

I further share reviewer 2's concerns about the use of a monthly integration time scale to assess causal links in phytoplankton systems when phytoplankton cells tend to have turnover times of 1-2 days. At temperate and high latitudes, blooms last weeks, not months, and species composition (and species richness) varies strongly throughout each bloom. Here again, I am not fully convinced by the authors' response with regard to this issue, and I would like them to repeat their analysis for each time series at the time scales that measurements are taken. This would allow for a comparison of the drivers and feedback strengths across the temporal scales available, and would allow for the quantification of the effect of temporal averaging.

Response:

We compared our original findings with the analysis based on non-integrated time series. As explained in our main text, applying EDM requires time series data arranged with equal time interval. Therefore, to compile monthly time series using non-integrated measurements while arranging data points with an equal interval, we designed two ways to generate monthly time series from those datasets with at least biweekly sampling frequency. Specifically, non-integrated data were chosen from either the early half ($\text{day} \leq 15$) or later half ($\text{day} > 15$) in a month, respectively. Ideally, this way of data compilation yields two independent monthly time series composed of non-integrated measurements if the field sampling was arranged very regularly with at least biweekly frequency. However, considering the missing data issue, we were able to carry such analysis from only eight systems, in which at least 80% of data points were available, including Lake Biwa (Bw), two stations in Lake Kasumigaura (Ks_3& Ks_9), Shin River (Sh), three stations in Lake Inba (Ib_a, Ib_b, & Ib_c), and Müggelsee (Mu). Other systems cannot produce comparable non-integrated time

series because biweekly or weekly data were not persistent throughout the sampling period (e.g., less frequent sampling in early stage of the monitoring).

Based on the analyses of the non-integrated time series (Fig. S13), we revealed the strong correlations between the causal strengths estimated by non-integrated datasets versus those estimated by monthly integrated datasets. This finding indicates that the estimates of causal strengths based on monthly integrated data are similar with the averaged estimates computed from the non-integrated time series. Therefore, our main findings are robust to data compilation (i.e., averaging) given the fixed time scale of analysis (determined by time intervals). We addressed this finding in Supplement (Fig. S13) but also caveated the problems of using integrated data as suggested by the Reviewer in Method at P22:L16- P23:L3, ”For consistency, monthly time series were generated by averaging over observations if sampling occurred on a finer timescale. Although such compilation potentially causes some inconsistency in smoothing temporal fluctuations of time series data among systems with various sampling frequencies, it was necessary because our methods based on state space reconstruction require time series data at equal intervals, dictating the temporal scale of analysis. In our case, monthly resolution is the only consensus that can be applied to all time series datasets and monthly average is the most representative measure at this scale. Nonetheless, causal strengths estimated by CCM analysis were robust to this data averaging according to our comparisons using eight stations where regular and frequent sampling (i.e., sampling frequency higher than monthly) were available (Fig. S13).”

Fig. S13. Estimated causal strength were robust to data integration that averages over the observations on monthly scale.

Comments:

My main concern, however, is that zooplankton are not included in this analysis. This is mentioned briefly in the caveats, but not in the detail I think is warranted here. In my view, this is a major concern and one of the key issues that may render some of the conclusions invalid. Zooplankton have long been monitored in lakes, and again I do not buy the explanation that these counts have been unavailable. In some of the 19 lakes perhaps, but we do not need 19 lakes to evaluate – and quantify – the effect of top-down pressures. Zooplankton grazing has been shown to be one of the primary drivers of phytoplankton diversity in many modeling studies, and top-down factors have also been identified a key determinant of lower trophic level network structure in recent global metagenomic studies. Due to the inverse biomass pyramid in marine and many freshwater systems, top-down pressure are much more important in aquatic systems than in terrestrial ones, and invasive species, such as the jellyfish *M. leidyi* have shown to change the phenological patterns of entire inland seas, such as the Caspian Sea. Hence, if there are causal links between diversity and biomass and nutrients, then those must not neglect the primary cause of phytoplankton demise, which is grazing by zooplankton. Zooplankton both store and integrate nutrient signals contained in the biomass of primary producers, and have also been found, and been suggested to control chlorophyll-a concentration in natural and modeled systems.

Response:

We agree with the Reviewer's concern about zooplankton. Unfortunately, the zooplankton data are not available for every system we analyzed. In some cases, zooplankton datasets, though available, have a much shorter time series length than the phytoplankton time series. Nonetheless, we did our best to collect as many zooplankton datasets as possible. Totally, we successfully collected long-term zooplankton time series from 11 systems (Table S3) and invited several new co-authors who are responsible for these zooplankton datasets. Our findings revealed the significant causal effects of herbivorous crustaceans on phytoplankton biomass and diversity in most of the analysed systems. However, zooplankton effects on phytoplankton biomass and diversity were on average not as strong as the effects of phytoplankton diversity and nitrates. Therefore, our main conclusion stating the importance of phytoplankton diversity effects remains robust to the inclusion of zooplankton analysis. Nonetheless, our findings with respect to zooplankton remain inconclusive because this analysis was not conducted in all 19 systems. Therefore, we only presented these findings in Supplement (Fig. S6) and addressed this finding in main text at P11:L7-19, "Apart from nutrients and temperature, the causal effects of

other important drivers on phytoplankton biomass and diversity were also examined, though not in all 19 systems due to data limitation. The causal effects of physical environmental factors, such as irradiance and water column stability, were presented in Fig. S5; the results indicated that the quantified causal strengths on average were not as strong as the effects of diversity and nutrients. Moreover, the effects of consumers (e.g., zooplankton), which have been suggested as important drivers affecting species diversity of phytoplankton communities⁴⁵, were also examined. Based on our analysis of zooplankton, causal effects of herbivorous crustaceans on phytoplankton biomass and diversity were significant in most of the analyzed systems. However, these effects were on average not as strong as the effects of phytoplankton diversity and nutrients, respectively (Fig. S6). Nonetheless, all these findings remained inconclusive, due to a lack of complete datasets across all systems (Table S3) and warrant more detailed investigation in future studies.”

Figure S6 | Causal strengths of zooplankton drivers on ecosystem functioning and species diversity in phytoplankton communities.

Comment:

The CCM method might be new, and attractive for the conceptual study of causal links in aquatic systems, but the authors also write that it is robust to moderate noise – whereas the noise in aquatic systems is high. Furthermore, any statistical algorithm will have difficulties in separating the influences of several highly correlated variables – in most marine systems, pretty much everything, from temperature to nutrient dynamics to mixing is correlated at larger spatio-temporal scales. Hence, great care

needs to be taken to avoid the identification of spurious links or methodological artifacts as patterns. The authors have thrown a powerful new method at their causality problem, but currently still fail to document their assumptions, and in particular the limits and limitations of their method in detail. What is the effect of each methodological choice taken, what is the uncertainty of this approach? While I am very excited about the approach, and think that this methodology has a lot of potential, I still fail to be convinced yet that this is not an example of a ‘garbage in – garbage out’ model. To convince me, I would need to see a more careful assessment of the method, variable selection (primary producer richness, standing stock, nutrients, temperature), the choice of proxies, etc. A sound sensitivity analysis, and solid uncertainty estimates where critical choices are being made would certainly convince me - and perhaps also reviewer 2.

Response:

CCM was developed based on Takens Theorem in dynamical system theory that relies on some assumptions about the inferred dynamical systems. In this revision, we added a new paragraph that included more detailed explanations about the assumptions and limitations of CCM in Methods at P24:L10-P25:L2, ”Several limitations in applying CCM analysis need to be acknowledged. First, CCM is based on lagged coordinated embedding in which each embedded variable needs to be lagged by a fixed time interval that determines the time scale of CCM analysis. For example, time series analyzed in our study were integrated to the monthly scale as stated in the *Data treatment* section. Multi-scale analysis is possible only when data points are measured very regularly across various time scales (e.g., from weekly to monthly). Second, as required in many time series analysis, CCM analysis requires time series data being stationary⁷⁶. Otherwise, CCM likely produces false positive findings (e.g., caused by strong seasonality); note however, we have removed seasonality in analysis in this work. Third, a time series including too many zero values (or other constant values) is not suitable for CCM analysis (as a general statistical issue in any time series analysis). This is because embedding such a time series potentially produces many zero vectors, which violates the general assumption of EDM that assumes a one-to-one mapping between each embedded vector and the vector on dynamical manifold⁷⁷ (i.e., zero vector can map to many possibilities on manifold). Thus, the embedded zero vectors need to be excluded or separated from the prediction set⁷³.” We also addressed how a non-parametric approach like CCM enables to consider the effects of confounding factors implicitly through embedding time lagged terms at P25:L3-9, ”CCM analysis accounts for influences of confounding variables implicitly. Specifically, CCM incorporates influences of confounding variables using lagged

embeddings, e.g., $(X_{t-1}, X_{t-2}, \dots)$, which have accounted for historical effects of other variables in lagged terms, even if those variables were unobserved or difficult to identify. As such, CCM does not require identifying or ruling out influences of confounding variables in order to quantify causations between two variables, and thus can be applied in more general dynamical systems⁶⁷.

To further improve the reliability of our statistical analysis, we firstly provided uncertainty estimates (i.e., standard errors) for all causal links and feedbacks (revised Fig. S3 and S7), which were computed from the sampling distribution reconstructed by 500 random resamples of embedded data points with replacement. This computation is based on the build-in options (resample=T and num_sample=500) in the 'ccm' function of rEDM package. The detailed R code was offered in Github, https://github.com/biozoo/Chang_etal_2021_SI_CausalFeedback. We also addressed the computation of standard errors in Method at P27:L17-20, “To determine the uncertainty of our estimates in causal strength and loop weight, we calculated their standard errors using resampling method that reconstructed sampling distributions from 500 random samples of embedded data points with replacement.”

In this response letter, we also provided supporting evidence of the reliability of our findings based on CCM. Firstly, we evaluated the inference quality of CCM by the CCM skills calculated as the correlation coefficients between empirical observations and model predictions. Then we compared the CCM skills with the results based on linear cross-correlation analysis that calculated the correlation coefficients between the time series of causal variable and lagged time series of effect variable (allowing at most three-month lags). Lastly, we evaluated the performance of cross-correlation analysis by the highest cross-correlation coefficients obtained from the analyses under various time lags. Compared to linear cross-correlation analysis, our CCM analysis, if significant, had reasonable inference quality in recovering empirical dynamics of key variables based on the inferred causal relationships. In most cases, CCM skill was higher than the optimal cross-correlation coefficient when a causal link was significant (i.e., converged in CCM). This finding indicates that the empirical dynamic modelling using CCM is not a ‘garbage in – garbage out’ model but faithfully recovered the empirical dynamics of analyzed variables. In fact, detailed method justifications of CCM have been revealed in many existing literatures (e.g., RefX6-9). Therefore, to avoid the redundancy, we only included this justification in response letter, with the understanding that the response letter will be published on-line. Nonetheless, we are willing to show our findings in Supplement if the Reviewer considers that necessary.

Fig. X1 CCM presented a reasonable high inference quality compared to linear cross-correlation analysis. For each variable pair with significant CCM causation, performance of CCM (cross-map skill) was compared with the best cross correlation. In a majority of variable pairs, cross-mapping skills of CCM were higher than the absolute values of cross correlation coefficients. A solid line indicates the 1:1 line. To ensure valid comparisons, cross-correlation allows a three-month lag response as in CCM; the lag with the best correlation coefficient was selected.

RefX6: Ye, H., E. R. Deyle, L. J. Gilarranz and G. Sugihara. (2015) Distinguishing time-delayed causal interactions using convergent cross mapping. *Scientific Reports* 5(1):14750.

RefX7: BozorgMagham, A. E., S. Motesharrei, S. G. Penny and E. Kalnay. (2015) Causality analysis: identifying the leading element in a coupled dynamical system. *PLoS ONE* 10:e0131226.

RefX8: Cummins, B., T. Gedeon and K. Spendlove. (2015) On the efficacy of state space reconstruction methods in determining causality. *SIAM Journal on Applied Dynamical Systems* 14:335-381.

RefX9: Chang, C. W., Ushio, M. & Hsieh, C.-h. (2017) Empirical dynamic modeling for beginners. *Ecological Research* 32, 785-796.

Comment:

To conclude, I do agree with reviewer 2 that a second round of review is warranted, but I don't agree that 'the data does not allow to answer the questions addressed'. This work has potential, and the method is exciting, and I would love to be convinced that what we see are actual signals.

Response:

Thank you for these constructive comments that really improve the reliability of our research. Based on these comments, we substantially revised the manuscript and included various sensitivity analyses to strengthen our main conclusions. As mentioned by the Reviewer, our current dataset, although not perfect, is still appropriate to address key questions about biodiversity-ecosystem functioning in aquatic systems.

Reviewers' Comments:

Reviewer #4:

Remarks to the Author:

I have had a careful look at the responses to issues raised by reviewer 2 and myself in the second round of the review process of this manuscript, and I am satisfied with the additional analysis conducted and the conclusions drawn based on these findings. I congratulate the authors on their very thorough responses - I have learnt a lot, and I believe the additional work was worthwhile and greatly improved the quality of the work. I am happy with the revision, and think this paper could be published without any need for a further round of reviews, if the editor and the other reviewers agree with this assessment.

I would just like the authors to change the following sentence:

"Nonetheless, all these findings remained inconclusive, due to a lack of complete datasets across all systems (Table S3) and warrant more detailed investigation in future studies."

I disagree with your assessment that these results are 'inconclusive'. In fact, you have shown convincingly with a lot of statistical analysis they are very conclusive - they just simply cannot be generalized to all 19 systems, since you have data for only 11 of them. Please revise this statement, as the current sentence does not do justice to the substantial amount of effort you invested into this additional analysis.

Reviewer #4 (Remarks to the Author):

I have had a careful look at the responses to issues raised by reviewer 2 and myself in the second round of the review process of this manuscript, and I am satisfied with the additional analysis conducted and the conclusions drawn based on these findings. I congratulate the authors on their very thorough responses - I have learnt a lot, and I believe the additional work was worthwhile and greatly improved the quality of the work. I am happy with the revision, and think this paper could be published without any need for a further round of reviews, if the editor and the other reviewers agree with this assessment. I would just like the authors to change the following sentence:

"Nonetheless, all these findings remained inconclusive, due to a lack of complete datasets across all systems (Table S3) and warrant more detailed investigation in future studies."

I disagree with your assessment that these results are 'inconclusive'. In fact, you have shown convincingly with a lot of statistical analysis they are very conclusive - they just simply cannot be generalized to all 19 systems, since you have data for only 11 of them. Please revise this statement, as the current sentence does not do justice to the substantial amount of effort you invested into this additional analysis.

Response:

We are very grateful for the insightful comments made by the Reviewer that substantially improve the reliability of our analyses. Nonetheless, we totally agree with the Reviewer that the sentence describing our additional analysis might downgrade the value of these new findings. Therefore, we only indicated a minor caveat that the analysis has not been generalized to all 19 systems and thus requires further investigation. We revised this sentence after presenting the findings of zooplankton analysis at P11:L7-9,

“Nonetheless, these findings were not generalized to all 19 systems due to a lack of complete datasets as shown in Table S3, and thus warrant more detailed investigation in future studies.”

Thank you very much for these constructive comments that critically improve the quality of our publication.